# Interpreting Distributional Reinforcement Learning: A Regularization Perspective

## Abstract

Distributional reinforcement learning (RL) is a class of state-of-the-art algorithms that estimate the entire distribution of the total return rather than its expected value alone. The theoretical advantages of distributional RL over expectation-based RL remain elusive, despite the remarkable performance of distributional RL. Our work attributes the potential superiority of distributional RL to its regularization effect stemming from the value distribution information regardless of only its expectation. We decompose the value distribution into its expectation and the remaining distribution part using a variant of the gross error model in robust statistics. Hence, distributional RL has an additional benefit over expectation-based RL thanks to the impact of a *risk-sensitive entropy regularization* within the Neural Fitted Z-Iteration framework. Meanwhile, we investigate the role of the resulting regularization in actor-critic algorithms by bridging the risk-sensitive entropy regularization of distributional RL and the vanilla entropy in maximum entropy RL. It reveals that distributional RL induces an augmented reward function, which promotes a risk-sensitive exploration against the intrinsic uncertainty of the environment. Finally, extensive experiments verify the importance of the regularization effect in distributional RL, as well as the mutual impacts of different entropy regularizations. Our study paves the way towards a better understanding of distributional RL, especially when looked at through a regularization lens.

## 1 Introduction

The intrinsic characteristics of classical reinforcement learning (RL) algorithms, such as temporal-difference (TD) learning (Sutton & Barto, 2018) and Q-learning (Watkins & Dayan, 1992), are based on the expectation of discounted cumulative rewards that an agent observes while interacting with the environment. In stark contrast to the classical expectation-based RL, a new branch of algorithms called distributional RL estimates the full distribution of total returns and has demonstrated the state-of-the-art performance in a wide range of environments (Bellemare et al., 2017a; Dabney et al., 2018b;a; Yang et al., 2019; Zhou et al., 2020; Nguyen et al., 2020; Sun et al., 2022). Meanwhile, distributional RL also inherits other benefits in risk-sensitive control (Dabney et al., 2018a), policy exploration settings (Mavrin et al., 2019; Rowland et al., 2019) and robustness (Sun et al., 2021).

Despite the existence of numerous algorithmic variants of distributional RL with remarkable empirical success, we still have a poor understanding of what the effectiveness of distributional RL is stemming from and theoretical studies on advantages of distributional RL over expectation-based RL are still less established. Distributional RL problems was also mapped to a Wasserstein gradient flow problem (Martin et al., 2020), treating the distributional Bellman residual as a potential energy functional. Offline distributional RL (Ma et al., 2021) has also been proposed to investigate the efficacy of distributional RL in both risk-neutral and risk-averse domains. (Lyle et al., 2019) proved in many realizations of tabular and linear approximation settings, distributional RL behaves the same as expectation-based RL under the coupling updates method, but diverges in non-linear approximation. Although the explanation from these works is not sufficient yet, the trend is encouraging for recent works towards closing the gap between theory and practice in distributional RL.

In this paper, we illuminate the behavior difference of distributional RL over expectation-based RL through the lens of regularization to explain its empirical outperformance in most practical environments. Specifically, we simplify distributional RL into a Neural Fitted Z-Iteration framework,

within which we establish an equivalence of objective functions between distributional RL and a risk-sensitive entropy regularized Neural Fitted Q-Iteration from the perspective of statistics. This result is based on two analytical components, i.e., action-value density function decomposition by leveraging of a variant of gross error model in robust statistics, as well as Kullback-Leibler (KL) divergence to measure the distribution distance between the current and target value distribution in each Bellman update. Then we establish a connection between the impact of risk-sensitive entropy regularization of distributional RL and vanilla entropy in maximum entropy RL, yielding a *Distribution-Entropy-Regularized Actor Critic* algorithm. Empirical results demonstrate the crucial role of risk-sensitive entropy regularization effect from distributional RL in the potential superiority over expectation-based RL on both Atari games and MuJoCo environments. We also reveal mutual impacts of both risk-sensitive entropy in distributional RL and vanilla entropy in maximum entropy RL, providing more potential research directions in the future.

## 2 PRELIMINARY KNOWLEDGE

In classical RL, an agent interacts with an environment via a Markov decision process (MDP), a 5-tuple $(\mathcal{S}, \mathcal{A}, R, P, \gamma)$, where $\mathcal{S}$ and $\mathcal{A}$ are the state and action spaces, respectively. $P$ is the environment transition dynamics, $R$ is the reward function and $\gamma \in (0, 1)$ is the discount factor.

**Action-value Function vs Action-value Distribution.** Given a policy $\pi$, the discounted sum of future rewards is a random variable $Z^\pi(s, a) = \sum_{t=0}^\infty \gamma^t R(s_t, a_t)$, where $s_0 = s$, $a_0 = a$, $s_{t+1} \sim P(\cdot|s_t, a_t)$, and $a_t \sim \pi(\cdot|s_t)$. In the control setting, expectation-based RL focuses on the action-value function $Q^\pi(s, a)$, the expectation of $Z^\pi(s, a)$, i.e., $Q^\pi(s, a) = \mathbb{E}[Z^\pi(s, a)]$. Distributional RL, on the other hand, focuses on the action-value distribution, the full distribution of $Z^\pi(s, a)$. The density function if exists of action-value distribution is called *action-value density function*.

**Bellman Operators vs Distributional Bellman Operators.** For the policy evaluation in expectation-based RL, the value function is updated via the Bellman operator $\mathcal{T}^\pi Q(s, a) = \mathbb{E}[R(s, a)] + \gamma \mathbb{E}_{s' \sim p, a' \sim \pi}[Q(s', a')]$. We also define Bellman Optimality Operator $\mathcal{T}^{\mathrm{opt}} Q(s, a) = \mathbb{E}[R(s, a)] + \gamma \max_{a'} \mathbb{E}_{s' \sim p}[Q(s', a')]$. In distributional RL, the action-value distribution of $Z^\pi(s, a)$ is updated via the distributional Bellman operator $\mathfrak{T}^\pi$, i.e., $\mathfrak{T}^\pi Z(s, a) \stackrel{D}{=} R(s, a) + \gamma Z(s', a')$, where $s' \sim P(\cdot|s, a)$ and $a' \sim \pi(\cdot|s')$. The equality implies that random variables of both sides are equal in distribution. This random-variable definition of distributional Bellman operator is appealing and easily understood due to its concise form, although its value-distribution definition is more mathematically rigorous (Rowland et al., 2018; Bellemare et al., 2022).

**Categorical Distributional RL.** Categorical Distributional RL (Bellemare et al., 2017a) can be viewed as the first successful distributional RL algorithm family that approximates the value distribution $\eta$ by a discrete categorical distribution $\hat{\eta} = \sum_{i=1}^N p_i \delta_{z_i}$, where $z_1 \leq z_2 \leq ... \leq z_N$ is a set of fixed supports and $\{p_i\}_{i=1}^N$ are learnable probabilities. The leverage of a heuristic projection operator $\Pi_{\mathcal{C}}$ (see Appendix A for more details) as well as the KL divergence allows the theoretical convergence of categorical distribution RL under Cramér distance (Rowland et al., 2018).

## 3 REGULARIZATION EFFECT OF DISTRIBUTIONAL RL

### 3.1 DISTRIBUTIONAL RL: NEURAL FITTED Z-ITERATION (NEURAL FZI)

**Expectation-based RL: Neural Fitted Q-Iteration (Neural FQI).** Neural FQI (Fan et al., 2020; Riedmiller, 2005) offers a statistical explanation of DQN (Mnih et al., 2015), capturing its key features, including experience replay and the target network $Q_{\theta^*}$. In Neural FQI, we update parameterized $Q_\theta(s, a)$ in each iteration $k$ in a regression problem:

$$Q_\theta^{k+1} = \operatorname*{argmin}_{Q_\theta} \frac{1}{n} \sum_{i=1}^n \left[y_i - Q_\theta^k(s_i, a_i)\right]^2,\tag{1}$$

where the target $y_i = r(s_i, a_i) + \gamma \max_{a \in \mathcal{A}} Q_{\theta^*}^k(s_i', a)$ is fixed within every $T_{\mathrm{target}}$ steps to update target network $Q_{\theta^*}$ by letting $\theta^* = \theta$. The experience buffer induces independent samples $\{(s_i, a_i, r_i, s_i')\}_{i \in [n]}$. In an ideal case when we neglect the non-convexity and TD approximation

errors, we have $Q_\theta^{k+1} = \mathcal{T}^{\text{opt}} Q_{\theta^*}^k$, which is exactly the updating rule under Bellman optimality operator (Fan et al., 2020). In the viewpoint of statistics, the optimization problem in Eq. 1 in each iteration is a standard supervised and neural network parameterized regression regarding $Q_\theta$.

**Distributional RL: Neural Fitted Z-Iteration (Neural FZI).** We interpret distributional RL as a Neural Fitted Z-Iteration owing to the fact that this iteration is by far closest to the practical algorithms and more interpretable. Analogous to Neural FQI, we can simplify value-based distributional RL algorithms parameterized by $Z_\theta$ into a Neural Fitted Z-Iteration (Neural FZI) as

$$Z_\theta^{k+1} = \underset{Z_\theta}{\arg\min} \frac{1}{n} \sum_{i=1}^n d_p(Y_i, Z_\theta^k(s_i, a_i)), \tag{2}$$

where the target $Y_i = R(s_i, a_i) + \gamma Z_{\theta^*}^k(s_i', \pi_Z(s_i'))$ with the policy $\pi_Z$ following the greedy rule $\pi_Z(s_i') = \arg\max_{a'} \mathbb{E}\left[Z_{\theta^*}^k(s_i', a')\right]$ is fixed within every $T_{\text{target}}$ steps to update target network $Z_{\theta^*}$. $d_p$ is a divergence between two distributions. Notably, choices of representation for $Z_\theta$ and the metric $d_p$ are pivotal for the empirical success of distributional RL algorithms (Sun et al., 2022).

## 3.2 DISTRIBUTIONAL RL: ENTROPY-REGULARIZED NEURAL FQI

**Action-Value Density Function Decomposition.** To separate the impact of additional distribution information from the expectation of $Z^\pi$, we leverage a variant of *gross error model* from robust statistics (Huber, 2004), which was also similarly used to analyze Label Smoothing (Müller et al., 2019) and Knowledge Distillation (Hinton et al., 2015). Particularly, we utilize a *histgram* $\widehat{p}^{s,a}(x)$ with $N$ bins to approximate an arbitrary continuous action-value density function $p^{s,a}(x)$ given a state $s$ and action $a$ as the histogram is probably the simplest approach for the density function estimate in the literature of non-parametric statistics. Given a fixed set of supports $z_0 \leq z_1 \leq ... \leq z_N$ with the equal bin size as $\Delta$, $\Delta_i = [z_{i-1}, z_i)$, $i = 1, ..., N-1$ with $\Delta_N = [z_{N-1}, z_N]$, the continuous histogram density function is $\widehat{p}^{s,a}(x) = \sum_{i=1}^N p_i \mathbb{1}(x \in \Delta_i)/\Delta$. Denote $\Delta_E$ as the interval that $\mathbb{E}\left[Z^\pi(s, a)\right]$ falls into, i.e., $\mathbb{E}\left[Z^\pi(s, a)\right] \in \Delta_E$. We conduct an action-value density function decomposition over $\widehat{p}^{s,a}(x)$ as follows:

$$\widehat{p}^{s,a}(x) = (1 - \epsilon)\mathbb{1}(x \in \Delta_E)/\Delta + \epsilon \sum_{i=1}^N p_i^\mu \mathbb{1}(x \in \Delta_i)/\Delta, \tag{3}$$

where $\widehat{p}^{s,a}$ induces a new histogram $\widehat{\mu}(x) = \sum_{i=1}^N p_i^\mu \mathbb{1}(x \in \Delta_i)/\Delta$ that is used to approximate a continuous density function $\mu(x)$. $\widehat{\mu}(x)$ or $\mu(x)$ aims at characterizing the impact of action-value distribution *regardless of* its expectation $\mathbb{E}\left[Z^\pi(s, a)\right]$ on the performance of distributional RL algorithms. $\epsilon$ controls the proportion between a single-bin histogram $\mathbb{1}(x \in \Delta_E)/\Delta$ and $\widehat{\mu}(x)$, where we will later show that this single-bin histogram function is linked to Neural FQI. Before diving deeper, we begin by showing that $\widehat{\mu}(x)$ is a valid density function under certain $\epsilon$ in Proposition 1.

**Proposition 1.** *Denote $\widehat{p}^{s,a}(x \in \Delta_E) = p_E/\Delta$. Following the density function decomposition in Eq. 3, $\widehat{\mu}(x) = \sum_{i=1}^N p_i^\mu \mathbb{1}(x \in \Delta_i)/\Delta$ is a valid probability density function $\iff \epsilon \geq 1 - p_E$.*

Proof is provided in Appendix B. We next show that the histogram density estimator $\widehat{p}^{s,a}(x)$ enjoys a uniform convergence rate to approximate an arbitrary continuous action-value density function $p^{s,a}(x)$ under the mild condition in Theorem 1. Proof is provided in Appendix C.

**Theorem 1.** *(Approximation Analysis of $\widehat{p}^{s,a}$) Suppose $p^{s,a}(x)$ is Lipschitz continuous and the support of a random variable is partitioned by N bins with bin size $\Delta$. Then*

$$\sup_x |\widehat{p}^{s,a}(x) - p^{s,a}(x)| = O(\Delta) + O_P\left(\sqrt{\frac{\log N}{n\Delta^2}}\right). \tag{4}$$

**Intermediate Role of Histogram Density Estimator.** Categorical and Quantile-based distributional RL are the two main families that estimate the true action-value density by the discrete categorical distribution and quantile function, respectively. We analyze that histogram density estimate plays an intermediate role within these two branches. *(1) Connection to Categorical Distributional RL.* Although the continuous histogram density estimator is in contrast to the discrete categorical distribution, in Proposition 2 (proof in Appendix D) we reveal that minimizing the KL divergence

between the target density function and the histogram density estimate is equivalent to the parameterized categorical distribution . *(2) Connection to Quantile-based Distributional RL.* Histogram and quantile functions are "two sides of a coin". The histogram estimates the density function by giving each bin an equal amount of information, while the quantile function gives each fraction of data the same amount of information. Based on this insight, we thus argue that our analysis based on the histogram density estimate is largely general and representative in distributional RL families.

**Proposition 2.** *(Equivalence to Categorical Distribution) Suppose the target categorical distribution $c = \sum_{i=1}^{N} p_i \delta_{z_i}$ and the target histogram function $h(x) = \sum_{i=1}^{N} p_i \mathbb{1}(x \in \Delta_i)/\Delta$, updating the parameterized categorical distribution $c_\theta$ under KL divergence is equivalent to updating the parameterized histogram function $h_\theta$.*

**Distributional RL: Entropy-regularized Neural FQI.** We apply the decomposition on the target action-value histogram density function and choose KL divergence as $d_p$ in Neural FZI. Let $\mathcal{H}(P, Q)$ be the cross entropy between two probability measures $P$ and $Q$, i.e., $\mathcal{H}(P, Q) = -\int_{x \in \mathcal{X}} P(x) \log Q(x) \, \mathrm{d}x$. The target histogram density function $\widehat{p}^{s,a}$ is decomposed as $\widehat{p}^{s,a}(x) = (1 - \epsilon)\mathbb{1}(x \in \Delta_E)/\Delta + \epsilon\widehat{\mu}(x)$. We can derive the following entropy-regularized form for distributional RL in Proposition 3.

**Proposition 3.** *Denote $q_\theta^{s,a}(x)$ as the histogram density function of $Z_\theta^k(s, a)$ in Neural FZI. Based on the decomposition in Eq. 3 and KL divergence as $d_p$, Neural FZI in Eq. 2 is simplified as*

$$Z_\theta^{k+1} = \operatorname*{argmin}_{q_\theta} \frac{1}{n} \sum_{i=1}^{n} \left[ -\log q_\theta^{s_i, a_i}(\Delta_E^i) + \alpha \mathcal{H}(\widehat{\mu}^{s_i', \pi_Z(s_i')}, q_\theta^{s_i, a_i}) \right], \tag{5}$$

where $\alpha = \varepsilon/(1 - \varepsilon) > 0$ and $\Delta_E^i$ represents the interval that $\mathbb{E}\left[Z^\pi(s_i', \pi_Z(s_i'))\right]$ falls into, i.e., $\mathbb{E}\left[Z^\pi(s_i', \pi_Z(s_i'))\right] \in \Delta_E^i$. $\widehat{\mu}^{s_i', \pi_Z(s_i')}$ is the resulting histogram density function in the next state action pair $(s_i', \pi_Z(s_i'))$. Proof is given in Appendix F. In Proposition 4 with proof in Appendix G, we further show that minimizing the first term in Eq. 5 is "almost" equivalent to minimizing Neural FQI . For the uniformity of notation, we still use $s, a$ in the following analysis instead of $s_i, a_i$.

**Proposition 4.** *(Connection between Neural FZI and FQI via Decomposition) In Eq. 5 of Neural FZI, if the function class $\{Z_\theta : \theta \in \Theta\}$ is sufficiently large such that it contains the target $\{Y_i\}_{i=1}^n$, where $Y_i = R(s_i, a_i) + \gamma Z_{\theta^*}^k(s_i', \pi_Z(s_i'))$. Minimizing **the first term** in Eq. 5, as $\Delta \to 0$ implies*

$$P(Z_\theta^{k+1}(s, a) = \mathcal{T}^{opt} Q_{\theta^*}^k(s, a)) = 1. \tag{6}$$

**Interpretation of Proposition 4.** Given the fact that $Q_\theta^{k+1} = \mathcal{T}^{\mathrm{opt}} Q_{\theta^*}^k$ ideally in Neural FQI (Fan et al., 2020), we have $Z_\theta^{k+1} = \mathcal{T}^{\mathrm{opt}} Q_{\theta^*}^k$ with probability one under the assumption in Proposition 4. This indicates $Z_\theta^{k+1}$ may take other values instead of its expectation part as $\mathbb{E}\left[Z_\theta^{k+1}\right] = \mathcal{T}^{\mathrm{opt}} Q_{\theta^*}^k$, but the probability when these events for other values happen is 0. This result establishes a theoretical link between Neural FZI regarding the first term in Eq. 5 with Neural FQI.

**Interpretation of Proposition 3.** Based on the equivalence between the first term of Neural FZI and FQI, we therefore interpret the distributional RL form in Eq. 5 as *entropy-regularized Neural FQI*. Thus, the second regularization term $\mathcal{H}(\widehat{\mu}^{s_i', \pi_Z(s_i')}, q_\theta^{s_i, a_i})$ aims at explaining the behavior difference between distributional RL and expectation-based RL. It pushes $q_\theta^{s,a}$ for the current state-action pair to approximate $\widehat{\mu}^{s_i', \pi_Z(s_i')}$ for the next state-action pair, which "deducts" the expectation effect from the whole action-value distribution by leveraging of the density function decomposition technique proposed in Eq. 3. In summary, we interpret impacts of these two terms in Eq. 5 on the distributional RL optimization as **expectation effect** and **distributional regularization effect**, respectively.

**Risk-Sensitive Entropy Regularization.** We attribute the behavior difference of distributional RL, especially the ability to significantly reduce intrinsic uncertainty of the environment (Mavrin et al., 2019), into the regularization term in Eq. 5. According to the literature of risks in RL (Dabney et al., 2018a), where "risk" refers to the uncertainty over possible outcomes and "risk-sensitive policies" are those which depend upon more than the mean of the outcomes, we hereby call the novel cross entropy regularization for the second term in Eq. 5 as *risk-sensitive entropy regularization*. This risk-sensitive entropy regularization derived within distributional RL expands the class of policies using information provided by the distribution over returns (i.e. to the class of risk-sensitive policies). It should also be noted that our risk-sensitive entropy regularization is indeed "risk-neutral"

in the sense of convexity or concaveness of utility functions, where our policy is still applying a linear utility function $U$, defined as $\pi(\cdot|s) = \arg\max_a \mathbb{E}_{Z(s,a)}[U(z)]$. Correspondingly, We can additionally vary different distortion risk measures to explicitly lead the policy to being risk-averse or risk-seeking (Dabney et al., 2018a).

**Remark on KL divergence.** As stated in categorical distributional RL in Section 2, when the categorical distribution is applied after the projection operator $\Pi_C$, distributional Bellman operator $\mathfrak{T}^\pi$ has the contraction guarantee under Cramér distance (Rowland et al., 2018), albeit the use of a non-expansive KL divergence (Morimura et al., 2012). Similarly, our histogram density function with the projection $\Pi_C$ equipped with KL divergence also enjoys a contraction property due to the equivalence between optimizing histogram function and categorical distribution analyzed in Proposition 2. We also summarize favorable properties of KL divergence in distributional RL in Appendix E.

**Remark on the Attainability of $\widehat{\mu}^{s',\pi_Z(s')}$.** In practical distributional RL algorithms, we typically use the bootstrap, e.g., TD learning, to attain the target probability density estimate $\widehat{\mu}^{s',\pi_Z(s')}$ based on Eq. 3 as long as $\mathbb{E}[Z(s,a)]$ exists and $\epsilon \geq 1 - p_E$ in Proposition 1. The leverage of $\widehat{\mu}^{s',\pi_Z(s')}$ and the regularization effect revealed in Eq. 5 of distributional RL de facto establishes a bridge with maximum entropy RL (Williams & Peng, 1991), on which we have a deeper analysis in Section 3.3.

### 3.3 CONNECTION WITH MAXIMUM ENTROPY RL

**Vanilla Entropy Regularization in Maximum Entropy RL.** Maximum entropy RL (Williams & Peng, 1991), including Soft Q-Learning (Haarnoja et al., 2017), explicitly optimizes for policies that aim to reach states where they will have high entropy in the future:

$$J(\pi) = \sum_{t=0}^{T} \mathbb{E}_{(s_t,a_t)\sim\rho_\pi}\left[r\left(s_t, a_t\right) + \beta\mathcal{H}(\pi(\cdot|s_t))\right], \tag{7}$$

where $\mathcal{H}\left(\pi_\theta\left(\cdot|s_t\right)\right) = -\sum_a \pi_\theta\left(a|s_t\right)\log\pi_\theta\left(a|s_t\right)$ and $\rho_\pi$ is the generated distribution following $\pi$. The temperature parameter $\beta$ determines the relative importance of the entropy term against the cumulative rewards, and thus controls the action diversity of the optimal policy learned via Eq. 7. This maximum entropy regularization has various conceptual and practical advantages. Firstly, the learned policy is encouraged to visit states with high entropy in the future, thus promoting the exploration over diverse states (Han & Sung, 2021). Secondly, it considerably improves the learning speed (Mei et al., 2020) and therefore is widely used in state-of-the-art algorithms, e.g., Soft Actor-Critic (SAC) (Haarnoja et al., 2018). Similar empirical benefits of both distributional RL and maximum entropy RL also encourage us to probe their underlying connection.

**Risk-Sensitive Entropy Regularization in Distributional RL.** To make a direct comparison with maximum entropy RL, we need to specifically analyze the impact of the regularization term in Eq. 5, and thus we incorporate the risk-sensitive entropy regularization of distributional RL into the policy gradient framework akin to maximum entropy RL. Concretely, we conduct our analysis by showing the convergence of *Distribution-Entropy-Regularized Policy Iteration* (DERPI), which is the counterpart for Soft Policy Iteration (Haarnoja et al., 2018), i.e., the underpinning of SAC algorithm. In principle, Distribution-Entropy-Regularized Policy Iteration replaces the vanilla entropy regularization in Soft Policy Iteration with our risk-sensitive entropy regularization in Eq. 5 from distributional RL. In the policy evaluation step of distribution-entropy-regularized policy iteration, a new soft Q-value, i.e., the expectation of $Z^\pi(s,a)$, can be computed iteratively by applying a modified Bellman operator $\mathcal{T}_d^\pi$, which we call *Distribution-Entropy-Regularized Bellman Operator* defined as

$$\mathcal{T}_d^\pi Q\left(s_t, a_t\right) \triangleq r\left(s_t, a_t\right) + \gamma\mathbb{E}_{s_{t+1}\sim P(\cdot|s_t,a_t)}\left[V\left(s_{t+1}|s_t, a_t\right)\right], \tag{8}$$

where a new soft value function $V\left(s_{t+1}|s_t, a_t\right)$ conditioned on $s_t, a_t$ is defined by

$$V\left(s_{t+1}|s_t, a_t\right) = \mathbb{E}_{a_{t+1}\sim\pi}\left[Q\left(s_{t+1}, a_{t+1}\right)\right] + f(\mathcal{H}\left(\mu^{s_t,a_t}, q_\theta^{s_t,a_t}\right)), \tag{9}$$

and $f$ is a continuous increasing function over the cross entropy $\mathcal{H}$. Note that in this specific tabular setting regarding $s_t$ and $a_t$, we particularly use $q_\theta^{s_t,a_t}(x)$ to approximate the true density function of $Z(s_t, a_t)$, and $\mu^{s_t,a_t}$ to represent the true target value distribution regardless of its expectation, which can normally be obtained via bootstrap estimate $\widehat{\mu}^{s_{t+1},\pi_Z(s_{t+1})}$ similar in Eq. 5. The $f$ transformation over the cross entropy $\mathcal{H}$ between $\mu^{s_t,a_t}$ and $q_\theta^{s_t,a_t}(x)$ serves as our *risk-sensitive entropy*

*regularization.* As opposed to the vanilla entropy regularization in maximum entropy RL that encourages the policy to explore, our risk-sensitive entropy regularization in distributional RL plays a role of *the reward correction* or *augmented reward*, and therefore augments the action-value function $Q(s_t, a_t)$ in the value-based RL and the objective function in policy gradient RL by additionally incorporating the value distribution knowledge. As we have discussed Neural FZI above in Section 3.2, which is established on the value-based RL, we now shift our attention to the properties of our risk-sensitive entropy regularization in the framework of policy gradient. In Lemma 1, we firstly show that our Distribution-Entropy-Regularized Bellman operator $\mathcal{T}_d^\pi$ still inherits the convergence property in the policy evaluation phase.

**Lemma 1.** *(Distribution-Entropy-Regularized Policy Evaluation) Consider the distribution-entropy-regularized Bellman operator $\mathcal{T}_d^\pi$ in Eq. 8 and the behavior of expectation of $Z^\pi(s, a)$, i.e., $Q(s, a)$. Assume $\mathcal{H}(\mu^{s_t, a_t}, q_\theta^{s_t, a_t}) \leq M$ for all $(s_t, a_t) \in \mathcal{S} \times \mathcal{A}$, where $M$ is a constant. Define $Q^{k+1} = \mathcal{T}_d^\pi Q^k$, then $Q^{k+1}$ will converge to a corrected Q-value of $\pi$ as $k \to \infty$ with the new objective function defined as*

$$J'(\pi) = \sum_{t=0}^{T} \mathbb{E}_{(s_t, a_t) \sim \rho_\pi} \left[ r(s_t, a_t) + \gamma f(\mathcal{H}(\mu^{s_t, a_t}, q_\theta^{s_t, a_t})) \right]. \tag{10}$$

In Lemma 1, we reveal that the new objective function for distributional RL can be interpreted as an augmented reward function. Secondly, in the policy improvement for distributional RL, we keep the vanilla policy improvement updating rules according to

$$\pi_{\text{new}} = \arg\max_{\pi' \in \Pi} \mathbb{E}_{a_t \sim \pi'} \left[ Q^{\pi_{\text{old}}}(s_t, a_t) \right]. \tag{11}$$

Next we can immediately derive a new policy iteration algorithm, called *Distribution-Entropy-Regularized Policy Iteration (DERPI)* that alternates between distribution-entropy-regularized policy evaluation in Eq. 8 and the policy improvement in Eq. 11. It will provably converge to the policy with the optimal risk-sensitive entropy among all policies in $\Pi$ as shown in Theorem 2.

**Theorem 2.** *(Distribution-Entropy-Regularized Policy Iteration) Assume $\mathcal{H}(\mu^{s_t, a_t}, q_\theta^{s_t, a_t}) \leq M$ for all $(s_t, a_t) \in \mathcal{S} \times \mathcal{A}$, where $M$ is a constant. Repeatedly applying distribution-entropy-regularized policy evaluation in Eq. 8 and the policy improvement in Eq. 11, the policy converges to an optimal policy $\pi^*$ such that $Q^{\pi^*}(s_t, a_t) \geq Q^\pi(s_t, a_t)$ for all $\pi \in \Pi$.*

Please refer to Appendix H for the proof of Lemma 1 and Theorem 2. According to Theorem 2, it turns out that if we incorporate the risk-sensitive entropy regularization into the policy gradient framework in Eq. 10, we are able to design a variant of "soft policy iteration" that can guarantee the convergence to an optimal policy. As such, we provide a comprehensive comparison between vanilla entropy in maximum entropy RL and risk-sensitive entropy in distributional RL as follows.

**Vanilla Entropy Regularization vs Risk-Sensitive Entropy Regularization. (1) Objective function.** By comparing two objective function $J(\pi)$ in Eq. 7 for maximum entropy RL and $J'(\pi)$ in Eq. 10 for distributional RL, distributional RL tries to maximize the risk-sensitive entropy regularization *w.r.t.* $\pi$. This indicates that the learned policy in distributional RL is encouraged to *visit state and action pairs in the future whose action-value distributions have a higher degree of dispersion, e.g., variance, in spite of its expectation*, thus promoting the **risk-sensitive exploration** to reduce the intrinsic uncertainty of the environment. An intuitive

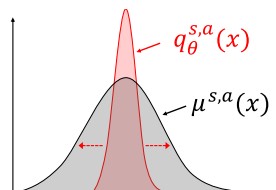

Figure 1: $q_\theta^{s,a}$ is encouraged to disperse under the risk-sensitive entropy regularization of distributional RL.

illustration is provided in Figure 1. **(2) State-action dependent regularization.** The vanilla entropy $\mathcal{H}(\pi(\cdot|s_t))$ in maximum entropy RL is state-wise, while our risk-sensitive regularization $\mathcal{H}(\mu^{s_t, a_t}, q_\theta^{s_t, a_t})$ is state-action-wise, implying that it is a more fine-grained regularization to characterize the action-value distribution of $Z(s_t, a_t)$ in the future.

## 3.4 ALGORITHM: DISTRIBUTION-ENTROPY-REGULARIZED ACTOR-CRITIC (DERAC)

In practice, large continuous domains require us to derive a practical approximation to DERPI. We thus extend DERPI from the tabular setting to the function approximation case, yielding the

Distribution-Entropy-Regularized Actor-Critic (DERAC) algorithm by using function approximators for both the value distribution $q_\theta(s_t, a_t)$ and the policy $\pi_\phi(a_t|s_t)$. The key characteristics of DERAC algorithm is that we use function approximator to represent the whole value distribution $q_\theta$ rather than only the value function, and conduct the optimization mainly based on the value function $Q_\theta(s_t, a_t) = \mathbb{E}[q_\theta(s_t, a_t)]$.

**Optimize the parameterized value distribution $q_\theta$.** The new value function is originally trained to minimize the squared residual error of Eq. 8. Here for a desirable interpretation, we impose the zero expectation assumption over the residual, i.e., $\mathcal{T}^\pi Q_\theta(s, a) = Q_\theta(s, a) + b$ with $\mathbb{E}[b] = 0$. The resulting simplified objective function $\hat{J}_q(\theta)$ can be well interpreted as an interpolation between the expectation effect and distributional regularization effect:

$$
\begin{aligned}
\hat{J}_q(\theta) &= \mathbb{E}_{s,a}\left[\left(\mathcal{T}_d^\pi Q_{\theta^*}(s, a) - Q_\theta(s, a)\right)^2\right] \\
&\propto (1-\lambda)\mathbb{E}_{s,a}\left[\left(\mathcal{T}^\pi \mathbb{E}[q_{\theta^*}(s, a)]\right] - \mathbb{E}[q_\theta(s, a)]\right)^2 + \lambda \mathbb{E}_{s,a}\left[\mathcal{H}(\mu^{s,a}, q_\theta^{s,a})\right],
\end{aligned}
\tag{12}
$$

where the results is simplified by using a particular increasing function $f(\mathcal{H}) = (\lambda\mathcal{H})^{\frac{1}{2}}/\gamma$ and $\lambda \in [0, 1]$ is the hyperparameter that controls the risk-sensitive regularization effect. Interestingly, when we leverage the whole target density function $\hat{p}^{s,a}$ to approximate the true $\mu^{s,a}$, the objective function in Eq. 12 can be viewed as an exact interpolation of loss functions between expectation-based RL (the first term) and categorical distributional RL loss (the second term), e.g., C51. Note that for the target $\mathcal{T}^\pi \mathbb{E}[q_{\theta^*}(s, a)]$, we use the target value distribution neural network $q_{\theta^*}$ to stabilize the training, which is consistent with the Neural FZI framework analyzed in Section 3.1.

**Optimize the policy $\pi_\phi$.** We optimize $\pi_\phi$ in Eq. 11 based on the $Q(s, a)$ and thus the new objective function $\hat{J}_\pi(\phi)$ can be expressed as $\hat{J}_\pi(\phi) = \mathbb{E}_{s,a\sim\pi_\phi}[\mathbb{E}[q_\theta(s, a)]]$. The complete DERAC algorithm is described in Algorithm 1 of Appendix J.

## 4 EXPERIMENTS

In the experiment, we firstly verify the regularization effect of distributional RL analyzed in Section 3.2 by decomposing the action-value histogram density function via Eq. 5 on both Atari games and MuJoCo environments. Next, we demonstrate the convergence and favorable performance of DERAC algorithm on continual control environments. Finally, an empirical extension to Implicit Quantile Networks (IQN) is provided to reveal mutual impacts of different entropy regularizations.

**Environments.** To demonstrate the value distribution decomposition, we mainly present results on three Atari games, including Breakout, Seaquest, Hero, over 3 seeds and three continuous control MuJoCo environments in OpenAI Gym, including ant, swimmer and bipedalwalkerhardcore, over 5 seeds. For the extension to IQN, we perform experiments on eight MuJoCo environments.

**Baselines.** To evaluate the risk-sensitive entropy regularization effect of distributional RL, we conduct an ablation study on C51 (Bellemare et al., 2017a) on Atari games and distributional SAC (DSAC) (Ma et al., 2020) on MuJoCo environments. The implementation of DERAC algorithm is based on distributional SAC (Haarnoja et al., 2018; Ma et al., 2020). More implementation details are provided in Appendix I.

### 4.1 DISTRIBUTION REGULARIZATION EFFECT OF DISTRIBUTIONAL RL

We demonstrate the rationale of action-value density function decomposition in Eq. 3 and the distribution regularization effect analyzed in Eq. 5 based on C51 algorithm equipped with KL divergence. Firstly, it is a fact that the value distribution decomposition is based on the equivalence between KL divergence and cross entropy owing to the leverage of target network. Hence, we demonstrate that C51 algorithm can still achieve similar results under the cross entropy loss across four Atari games in Figure 5 of Appendix K. In both the value-based C51 loss and the critic loss in DSAC with C51, we replace the whole target categorical distribution $\hat{p}^{s,a}(x)$ in C51 with the derived $\hat{\mu}^{s,a}(x)$ under different $\varepsilon$ in the cross entropy loss, allowing to investigate the risk-sensitive regularization effect of distributional RL. Concretely, we define $\varepsilon$ as the proportion of probability of the bin that contains the expectation *with the mass to transport to other bins*. We use $\varepsilon$ to replace $\epsilon$ for convenience as the leverage of $\varepsilon$ can always guarantee the valid density function $\hat{\mu}$ analyzed in Proposition 1. A large proportion probability $\varepsilon$ that transports less mass to other bins, corresponds to a large $\epsilon$ in Eq. 3.

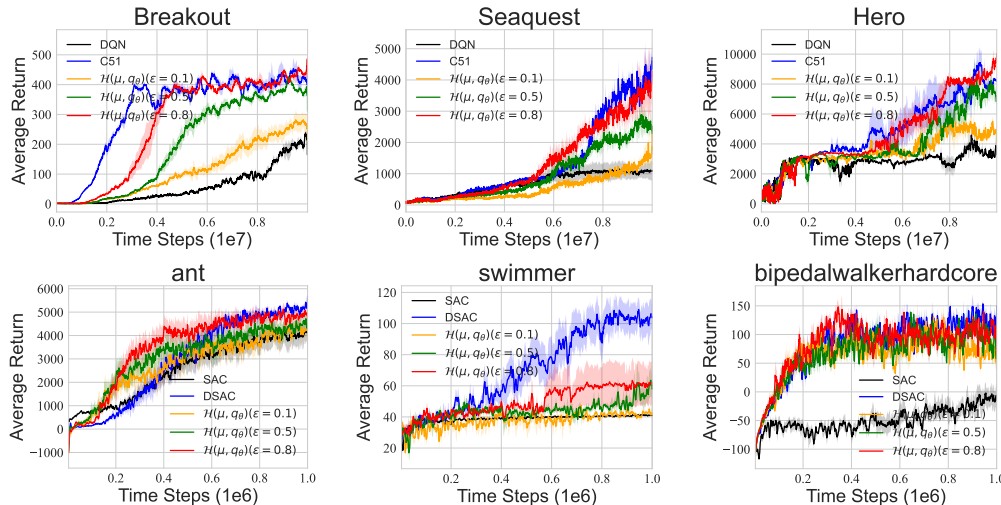

Figure 2: (**First Row**) Learning curves of C51 with value distribution decomposition $\mathcal{H}(\mu, q_\theta)$ under different $\varepsilon$ on three Atari games over 3 seeds. (**Second Row**) Learning curves of C51 with value distribution decomposition $\mathcal{H}(\mu, q_\theta)$ under different $\varepsilon$ on three MuJoCo environments over 5 seeds.

As shown in Figure 2, when $\varepsilon$ gradually decreases from 0.8 to 0.1, the learning curves of C51 $\mathcal{H}(\mu, q_\theta)$ tend to degrade from vanilla C51 to DQN across both Atari and MuJoCo, although their sensitivity in terms of $\varepsilon$ may depend on the environment, e.g., bipedalwalkerhardcore. This empirical observation corroborates the theoretical results we derive in Section 3.2, suggesting that risk-sensitive entropy regularization is pivotal to the success of distributional RL algorithms.

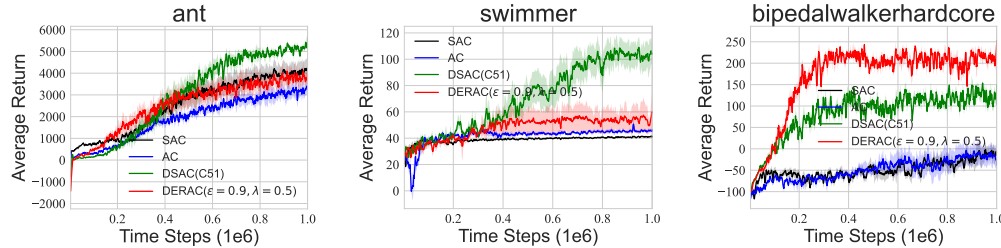

Figure 3: Learning curves of DERAC algorithms on three MuJoCo environments over 5 seeds.

## 4.2 CONVERGENCE OF DERAC ALGORITHM

We further demonstrate the convergence of DERAC algorithm. Figure 3 showcases that DERAC converges and achieves desirable performance on MuJoCo environments compared with AC (SAC without vanilla entropy) in the blue line. More importantly, Distribution-Entropy-Regularization (DER) in the red line could be remarkably beneficial for learning on the complex Bipedalwalkerhardcore, where a risk-sensitive exploration significantly improves the performance. It is worthwhile to know that our goal to introduce DERAC algorithm is not to pursue the empirical superiority of performance, but to corroborate the theoretical convergence of DERAC algorithm and DERPI in Theorem 2. In addition, as we choose $\varepsilon = 0.9$ in DERAC algorithm, there exists a distribution information loss, resulting in the learning performance degradation, e.g., on Swimmer. In practice, we can directly deploy distributional SAC to seek for a better performance. We also provide a sensitivity analysis of DERAC regarding $\lambda$ in Figure 6 of Appendix K.

## 4.3 EXTENSION TO QUANTILE-BASED DISTRIBUTIONAL RL

Finally, due to the fact that our aforementioned theoretical analysis is closely connected to categorical distributional RL algorithms, e.g., C51, in order to make a comprehensive conclusion in

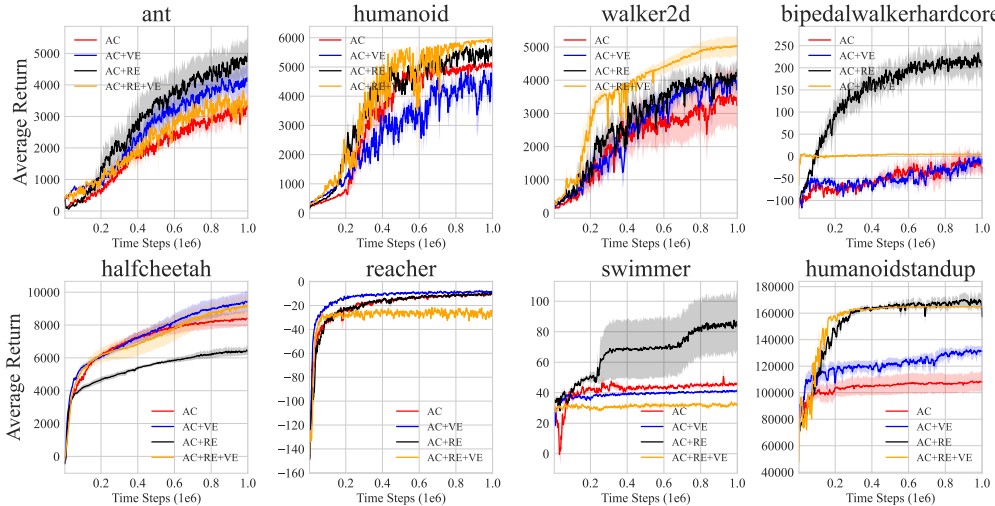

Figure 4: Learning curves of AC, AC+VE, AC+RE and AC+RE+VE over 5 seeds with smooth size 5 across eight MuJoCo environments where distributional RL part is based on IQN.

broader distributional RL branches, we thus heuristically extend our results in quantile-based distributional RL. This investigation is also based on the intermediate role of histogram function we analyzed in Section 3.2. Specifically, a careful ablation study is conducted to control the effects of vanilla entropy (VE), risk-sensitive entropy (RE) and their mutual impact. We denote SAC with and without vanilla entropy as *AC* and *AC+VE*, and distributional SAC with and without vanilla entropy as *AC+RE+VE* and *AC+RE*, where VE and RE are short for *Vanilla Entropy* and *Risk-sensitive Entropy*. For the implementation, we leverage the quantiles generation strategy in IQN (Dabney et al., 2018a) in distributional SAC (Ma et al., 2020). Hyper-parameters are listed in Appendix I. As suggested in Figure 4, although both vanilla entropy and risk-sensitive entropy effects may vary for different environments, we make the following conclusions:

**(1)** Vanilla entropy effect can enhance the performance as it is easily observed that AC+VE (blue line) outperforms AC (red lines) across most environments except on the humanoid and swimmer. The risk-sensitive entropy effect (RE) from distributional RL is also able to benefit the learning due to the fact that AC+RE (black lines) is more likely to bypass AC (red lines) especially on the complex BipealWalkerHardcore environment (hard for exploration).

**(2)** The use of both risk-sensitive entropy and vanilla entropy may interfere with each other, e.g., on BipealWalkerHardcore and Swimmer games, where *AC+RE+VE* (orange lines) is significantly inferior to *AC+RE* (black lines). This may results from the different exploration preference of two regularization effects. SAC encourages the policy to visit states with high entropy to pursue the diversity of states to optimize, while distributional RL promotes the risk-sensitive exploration to visit state and action pairs whose action-value distribution has larger degree of dispersion. We hypothesize that mixing two different exploration directions may lead to sub-optimal solutions in certain environments, thus interfering with each other eventually.

## 5 DISCUSSIONS AND CONCLUSION

Our regularization interpretation is based on histogram function equipped the KL divergence, strongly connected with categorical distributional RL. Although the histogram is linked with quantile function as well, a direct analysis based on quantile function is also promising in the future.

In this paper, we illuminate the behavior difference of distributional RL over expectation-based RL from the perspective of regularization. A risk-sensitive entropy regularization is derived for distributional RL within Neural FZI to explain the potential advantage of distributional RL. We also establish a connection between distributional RL with maximum entropy RL. Our research contributes to a deeper understanding of the potential superiority of distributional RL algorithms.

**Ethics Statement.** As our study is related to reveal the regularization effect of distributional RL algorithms, it is not involved with any ethics issue in our opinion.

**Reproducibility Statement.** Our results is based on the public implementation released in (Ma et al., 2020) with necessary implementation details given in Appendix I. We also provide the detailed proof in Appendix.

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

## A    Convergence Guarantee of Categorical Distributional RL

Categorical Distributional RL (Bellemare et al., 2017a) uses the heuristic projection operator $\Pi_{\mathcal{C}}$ that was defined as

$$\Pi_{\mathcal{C}}\left(\delta_y\right) = \begin{cases} \delta_{z_1} & y \leq z_1 \\ \frac{z_{i+1}-y}{z_{i+1}-z_i}\delta_{z_i} + \frac{y-z_i}{z_{i+1}-z_i}\delta_{z_{i+1}} & z_i < y \leq z_{i+1} \\ \delta_{z_K} & y > z_K \end{cases}, \tag{13}$$

and extended affinely to finite mixtures of Dirac measures, so that for a mixture of Diracs $\sum_{i=1}^{N} p_i \delta_{y_i}$, we have $\Pi_{\mathcal{C}}\left(\sum_{i=1}^{N} p_i \delta_{y_i}\right) = \sum_{i=1}^{N} p_i \Pi_{\mathcal{C}}\left(\delta_{y_i}\right)$. The Cramér distance was recently studied as an alternative to the Wasserstein distances in the context of generative models (Bellemare et al., 2017b). Recall the definition of Cramér distance.

**Definition 1.** *(Definition 3 (Rowland et al., 2018)) The Cramér distance $\ell_2$ between two distributions $\nu_1, \nu_2 \in \mathscr{P}(\mathbb{R})$, with cumulative distribution functions $F_{\nu_1}, F_{\nu_2}$ respectively, is defined by:*

$$\ell_2\left(\nu_1, \nu_2\right) = \left(\int_{\mathbb{R}} \left(F_{\nu_1}(x) - F_{\nu_2}(x)\right)^2 \, \mathrm{d}x\right)^{1/2}.$$

*Further, the supremum-Cramér metric $\bar{\ell}_2$ is defined between two distribution functions $\eta, \mu \in \mathscr{P}(\mathbb{R})^{\mathcal{X} \times \mathcal{A}}$ by*

$$\bar{\ell}_2(\eta, \mu) = \sup_{(x,a) \in \mathcal{X} \times \mathcal{A}} \ell_2\left(\eta^{(x,a)}, \mu^{(x,a)}\right).$$

Thus, the contraction of categorical distributional RL can be guaranteed under Cramér distance:

**Proposition 5.** *(Proposition 2 (Rowland et al., 2018)) The operator $\Pi_{\mathcal{C}}\mathcal{T}^\pi$ is a $\sqrt{\gamma}$-contraction in $\bar{\ell}_2$.*

An insight behind this conclusion is that Cramér distance endows a particular subset with a notion of orthogonal projection, and the orthogonal projection onto the subset is exactly the heuristic projection $\Pi_{\mathcal{C}}$ (Proposition 1 in (Rowland et al., 2018)).

## B    Proof of Proposition 1

**Proposition 1**. Denote $\widehat{p}^{s,a}(x \in \Delta_E) = p_E/\Delta$. Following the density function decomposition in Eq. 3, $\widehat{\mu}(x) = \sum_{i=1}^{N} p_i^\mu \mathbb{1}(x \in \Delta_i)/\Delta$ is a valid probability density function $\iff \epsilon \geq 1 - p_E$.

*Proof.* Recap a valid probability density function requires non-negative and one-bounded probability in each bin and all probabilities should sum to 1.

**Necessity.** (1) When $x \in \Delta_E$, Eq. 3 can simplified as $p_E/\Delta = (1 - \epsilon)/\Delta + \epsilon p_E^\mu/\Delta$, where $p_E^\mu = \widehat{\mu}(x \in \Delta_E)$. Thus, $p_E^\mu = \frac{p_E}{\epsilon} - \frac{1-\epsilon}{\epsilon} \geq 0$ if $\epsilon \geq 1 - p_E$. Obviously, $p_E^\mu = \frac{p_E}{\epsilon} - \frac{1-\epsilon}{\epsilon} \leq \frac{1}{\epsilon} - \frac{1-\epsilon}{\epsilon} = 1$ guaranteed by the validity of $\widehat{p}_E^{s,a}$. (2) When $x \notin \Delta_E$, we have $p_i/\Delta = \epsilon p_i^\mu/\Delta$, i.e.,When $x \notin \Delta_E$, We immediately have $p_i^\mu = \frac{p_i}{\epsilon} \leq \frac{1-p_E}{\epsilon} \leq 1$ when $\epsilon \geq 1 - p_E$. Also, $p_i^\mu = \frac{p_i}{\epsilon} \geq 0$.

**Sufficiency.** (1) When $x \in \Delta_E$, let $p_E^\mu = \frac{p_E}{\epsilon} - \frac{1-\epsilon}{\epsilon} \geq 0$, we have $\epsilon \geq 1 - p_E$. $p_E^\mu = \frac{p_E}{\epsilon} - \frac{1-\epsilon}{\epsilon} \leq 1$ in nature. (2) When $x \notin \Delta_E$, $p_i^\mu = \frac{p_i}{\epsilon} \geq 0$ in nature. Let $p_i^\mu = \frac{p_i}{\epsilon} \leq 1$, we have $p_i \leq \epsilon$. We need to take the intersection set of (1) and (2), we find that $\epsilon \geq 1 - p_E \Rightarrow \epsilon \geq 1 - p_E \geq p_i$ that satisfies condition in (2). Thus, the intersection set of (1) and (2) would be $\epsilon \geq 1 - p_E$.

In summary, as $\epsilon \geq 1 - p_E$ is both the necessary and sufficient condition, we have the conclusion that $\widehat{\mu}(x)$ is a valid probability density function $\iff \epsilon \geq 1 - p_E$.

$\square$

## C  PROOF OF THEOREM 1

**Theorem 1**. Suppose $p^{s,a}(x)$ is Lipschitz continuous and the support of $X$ is partitioned by N bins with bin size $\Delta$. Then

$$\sup_x |\widehat{p}^{s,a}(x) - p^{s,a}(x)| = O(\Delta) + O_P\left(\sqrt{\frac{\log N}{n\Delta^2}}\right). \tag{14}$$

*Proof.* Our proof mainly refers to (Wasserman, 2006). In particular, the difference of $\widehat{p}^{s,a}(x) - p^{s,a}(x)$ can be written as

$$\widehat{p}^{s,a}(x) - p^{s,a}(x) = \underbrace{\mathbb{E}\left(\widehat{p}^{s,a}(x)\right) - p(x)}_{\text{bias}} + \underbrace{\widehat{p}^{s,a}(x) - \mathbb{E}\left(\widehat{p}^{s,a}(x)\right)}_{\text{stochastic variation}}. \tag{15}$$

**(1) The first bias term.** Without loss of generality, we consider $x \in \Delta_k$, we have

$$\begin{aligned}
\mathbb{E}\left(\widehat{p}^{s,a}(x)\right) &= \frac{P(X \in \Delta_k)}{\Delta} \\
&= \frac{\int_{z_0+(k-1)\Delta}^{z_0+k\Delta} p(y)dy}{\Delta} \\
&= \frac{F(z_0 + (k-1)\Delta) - F(z_0 + (k-1)\Delta)}{z_0 + k\Delta - (z_0 + (k-1)\Delta)} \\
&= p^{s,a}(x'),
\end{aligned} \tag{16}$$

where the last equality is based on the mean value theorem. According the L-Lipschitz continuity property, we have

$$\begin{aligned}
|\mathbb{E}\left(\widehat{p}^{s,a}(x)\right) - p^{s,a}(x)| &= |p^{s,a}(x') - p^{s,a}(x)| \\
&\leq L|x' - x| \\
&\leq L\Delta
\end{aligned} \tag{17}$$

**(2) The second stochastic variation term.** If we let $x \in \Delta_k$, then $\widehat{p}^{s,a} = p_k = \frac{1}{n}\sum_{i=1}^n \mathbb{1}(X_i \in \Delta_k)$, we thus have

$$\begin{aligned}
&P\left(\sup_x |\widehat{p}^{s,a}(x) - \mathbb{E}\left(\widehat{p}^{s,a}(x)\right)| > \epsilon\right) \\
&= P\left(\max_{j=1,\cdots,N} \left|\frac{1}{n}\sum_{i=1}^n \mathbb{1}(X_i \in \Delta_j)/\Delta - P(X_i \in \Delta_j)/\Delta\right| > \epsilon\right) \\
&= P\left(\max_{j=1,\cdots,N} \left|\frac{1}{n}\sum_{i=1}^n \mathbb{1}(X_i \in \Delta_j) - P(X_i \in \Delta_j)\right| > \Delta\epsilon\right) \\
&\leq \sum_{j=1}^N P\left(\left|\frac{1}{n}\sum_{i=1}^n \mathbb{1}(X_i \in \Delta_j) - P(X_i \in \Delta_j)\right| > \Delta\epsilon\right) \\
&\leq N \cdot \exp\left(-2n\Delta^2\epsilon^2\right) \quad \text{(by Hoeffding's inequality)},
\end{aligned} \tag{18}$$

where in the last inequality we know that the indicator function is bounded in [0, 1]. We then let the last term be a constant independent of $N, n, \Delta$, thus,

$$\sup_x |\widehat{p}^{s,a}(x) - \mathbb{E}\left(\widehat{p}^{s,a}(x)\right)| = O_P\left(\sqrt{\frac{\log N}{n\Delta^2}}\right) \tag{19}$$

In summary, as the above inequality holds for each $x$, we thus have the uniform convergence rate of a histogram density estimator

$$\begin{aligned}
\sup_x |\widehat{p}^{s,a}(x) - p^{s,a}(x)| &\leq \sup_x |\mathbb{E}\left(\widehat{p}^{s,a}(x)\right) - p^{s,a}(x)| + \sup_x |\widehat{p}^{s,a}(x) - \mathbb{E}\left(\widehat{p}^{s,a}(x)\right)| \\
&= O(\Delta) + O_P\left(\sqrt{\frac{\log N}{n\Delta^2}}\right).
\end{aligned} \tag{20}$$

$\square$

## D    PROOF OF PROPOSITION 2

**Proposition 2**. Suppose the target categorical distribution $c = \sum_{i=1}^{N} p_i \delta_{z_i}$ and the target histogram function $h(x) = \sum_{i=1}^{N} p_i \mathbb{1}(x \in \Delta_i)/\Delta$, updating the parameterized categorical distribution $c_\theta$ under KL divergence is equivalent to updating the parameterized histogram function $h_\theta$.

*Proof.* For the histogram density estimator $h_\theta$ and the true target density function $p(x)$, we can simplify the KL divergence as follows.

$$
\begin{aligned}
D_{\mathrm{KL}}(h, h_\theta) &= \sum_{i=1}^{N} \int_{z_{i-1}}^{z_i} \frac{p_i(x)}{\Delta} \log \frac{\frac{p_i(x)}{\Delta}}{\frac{h_\theta^i}{\Delta}} dx \\
&= \sum_{i=1}^{N} \int_{z_{i-1}}^{z_i} \frac{p_i(x)}{\Delta} \log \frac{p_i(x)}{\Delta} dx - \sum_{i=1}^{N} \int_{z_{i-1}}^{z_i} \frac{p_i(x)}{\Delta} \log \frac{h_\theta^i}{\Delta} dx \\
&\propto -\sum_{i=1}^{N} \int_{z_{i-1}}^{z_i} \frac{p_i(x)}{\Delta} \log \frac{h_\theta^i}{\Delta} dx \\
&= -\sum_{i=1}^{N} p_i(x) \log \frac{h_\theta^i}{\Delta} \propto -\sum_{i=1}^{N} p_i(x) \log h_\theta^i
\end{aligned}
\tag{21}
$$

where $h_\theta^i$ is determined by $i$ and $\theta$ and is independent of $x$. For categorical distribution estimator $c_\theta$ with the probability $p_i$ in for each atom $z_i$, we also have its target categorical distribution $p(x)$ with each probability $p_i$, we have:

$$
\begin{aligned}
D_{\mathrm{KL}}(c, c_\theta) &= \sum_{i=1}^{N} p_i \log \frac{p_i}{c_\theta^i} \\
&= \sum_{i=1}^{N} p_i \log p_i - \sum_{i=1}^{N} p_i \log c_\theta^i \\
&\propto -\sum_{i=1}^{N} p_i \log c_\theta^i
\end{aligned}
\tag{22}
$$

$\square$

In categorical distributional RL we only use a discrete categorical distribution with probabilities centered on the fixed atoms $\{z_i\}_{i=1}^{N}$, while the histogram density estimator in our analysis is a continuous function defined on $[z_0, z_N]$. We reveal that minimizing the KL divergence regarding the parameterized categorical distribution in Eq. 22 is equivalent to minimizing the cross entropy loss regarding the parameterized histogram function in Eq. 21.

## E    PROPERTIES OF KL DIVERGENCE IN DISTRIBUTIONAL RL

**Proposition 6.** *Given two probability measures $\mu$ and $\nu$, we define the supreme $D_{KL}$ as a functional $\mathcal{P}(\mathcal{X})^{\mathcal{S} \times \mathcal{A}} \times \mathcal{P}(\mathcal{X})^{\mathcal{S} \times \mathcal{A}} \to \mathbb{R}$, i.e., $D_{KL}^\infty(\mu, \nu) = \sup_{(x,a) \in \mathcal{S} \times \mathcal{A}} D_{KL}(\mu(x,a), \nu(x,a))$. we have: (1) $\mathfrak{T}^\pi$ is a non-expansive distributional Bellman operator under $D_{KL}^\infty$, i.e., $D_{KL}^\infty(\mathfrak{T}^\pi Z_1, \mathfrak{T}^\pi Z_2) \leq D_{KL}^\infty(Z_1, Z_2)$, (2) $D_{KL}^\infty(Z_n, Z) \to 0$ implies the Wasserstein distance $W_p(Z_n, Z) \to 0$, (3) the expectation of $Z^\pi$ is still $\gamma$-contractive under $D_{KL}^\infty$, i.e., $\|\mathbb{E}\mathfrak{T}^\pi Z_1 - \mathbb{E}\mathfrak{T}^\pi Z_2\|_\infty \leq \gamma \|\mathbb{E}Z_1 - \mathbb{E}Z_2\|_\infty$.*

*Proof.* We firstly assume $Z_\theta$ is absolutely continuous and the supports of two distributions in KL divergence have a negligible intersection (Arjovsky & Bottou, 2017), under which the KL divergence is well-defined.

(1) Please refer to (Morimura et al., 2012) for the proof. Therefore, we have $D_{\mathrm{KL}}^\infty(\mathfrak{T}^\pi Z_1, \mathfrak{T}^\pi Z_2) \leq D_{\mathrm{KL}}^\infty(Z_1, Z_2)$, implying that $\mathfrak{T}^\pi$ is a non-expansive operator under $D_{\mathrm{KL}}^\infty$.

(2) By the definition of $D_{\mathrm{KL}}^\infty$, we have $\sup_{s,a} D_{\mathrm{KL}}(Z_n(s,a), Z(s,a)) \to 0$ implies $D_{\mathrm{KL}}(Z_n, Z) \to 0$. $D_{\mathrm{KL}}(Z_n, Z) \to 0$ implies the total variation distance $\delta(Z_n, Z) \to 0$ according to a straightforward application of Pinsker's inequality

$$
\begin{aligned}
\delta(Z_n, Z) &\leq \sqrt{\frac{1}{2} D_{\mathrm{KL}}(Z_n, Z)} \to 0 \\
\delta(Z, Z_n) &\leq \sqrt{\frac{1}{2} D_{\mathrm{KL}}(Z, Z_n)} \to 0
\end{aligned}
\tag{23}
$$

Based on Theorem 2 in WGAN (Arjovsky et al., 2017), $\delta(Z_n, Z) \to 0$ implies $W_p(Z_n, Z) \to 0$. This is trivial by recalling the fact that $\delta$ and $W$ give the strong an weak topologies on the dual of $(C(\mathcal{X}), \|\cdot\|_\infty)$ when restricted to $\mathrm{Prob}(\mathcal{X})$.

(3) The conclusion holds because the $\mathfrak{T}^\pi$ degenerates to $\mathcal{T}^\pi$ regardless of the metric $d_p$ (Bellemare et al., 2017a). Specifically, due to the linearity of expectation, we obtain that

$$
\|\mathbb{E}\mathfrak{T}^\pi Z_1 - \mathbb{E}\mathfrak{T}^\pi Z_2\|_\infty = \|\mathcal{T}^\pi \mathbb{E}Z_1 - \mathcal{T}^\pi \mathbb{E}Z_2\|_\infty \leq \gamma\|\mathbb{E}Z_1 - \mathbb{E}Z_2\|_\infty.
\tag{24}
$$

This implies that the expectation of $Z$ under $D_{\mathrm{KL}}$ exponentially converges to the expectation of $Z^*$, i.e., $\gamma$-contraction. $\qquad\square$

## F PROOF OF PROPOSITION 3

**Proposition 3** Denote $q_\theta^{s,a}(x)$ as the histogram density function of $Z_\theta^k(s,a)$ in Neural FZI. Based on the decomposition in Eq. 3 and KL divergence as $d_p$, Neural FZI in Eq. 2 is simplified as

$$
Z_\theta^{k+1} = \underset{q_\theta}{\arg\min} \frac{1}{n} \sum_{i=1}^{n} \left[ -\log q_\theta^{s_i, a_i}(\Delta_E^i) + \alpha \mathcal{H}(\widehat{\mu}^{s_i', \pi_Z(s_i')}, q_\theta^{s_i, a_i}) \right],
\tag{25}
$$

*Proof.* Firstly, given a fixed $p(x)$ we know that minimizing $D_{\mathrm{KL}}(p, q_\theta)$ is equivalent to minimizing $\mathcal{H}(p, q)$ by following

$$
\begin{aligned}
D_{\mathrm{KL}}(p, q_\theta) &= \sum_{i=1}^{N} \int_{z_{i-1}}^{z_i} p_i(x)/\Delta \log \frac{p^i(x)/\Delta}{q_\theta^i/\Delta} \, \mathrm{d}x \\
&= -\sum_{i=1}^{N} \int_{z_{i-1}}^{z_i} p_i(x)/\Delta \log q_\theta^i/\Delta \, \mathrm{d}x - \left(\sum_{i=1}^{N} \int_{z_{i-1}}^{z_i} p_i(x)/\Delta \log p^i(x)/\Delta \, \mathrm{d}x\right) \\
&= \mathcal{H}(p, q_\theta) - \mathcal{H}(p) \\
&\propto \mathcal{H}(p, q_\theta)
\end{aligned}
\tag{26}
$$

Based on $\mathcal{H}(p, q_\theta)$, we use $p^{s_i', \pi_Z(s_i')}(x)$ to denote the target probability density function of the random variable $R(s_i, a_i) + \gamma Z_{\theta^*}^k(s_i', \pi_Z(s_i'))$. Then, we can derive the objective function within

each Neural FZI as

$$\frac{1}{n}\sum_{i=1}^{n}\mathcal{H}(p^{s'_i,\pi_Z(s'_i)}(x), q_\theta^{s_i,a_i})$$

$$=\frac{1}{n}\sum_{i=1}^{n}\left((1-\epsilon)\mathcal{H}(\mathbb{1}(x\in\Delta_E^i)/\Delta, q_\theta^{s_i,a_i}/\Delta)+\epsilon\mathcal{H}(\widehat{\mu}^{s'_i,\pi_Z(s'_i)}/\Delta, q_\theta^{s_i,a_i}/\Delta)\right)$$

$$=\frac{1}{n}\sum_{i=1}^{n}\left(-(1-\epsilon)\sum_{j=1}^{N}\int_{z_{j-1}}^{z_j}\mathbb{1}(x\in\Delta_E^i)/\Delta\log q_\theta^{s_i,a_i}(\Delta_j)/\Delta dx-\epsilon\sum_{j=1}^{N}\int_{z_{j-1}}^{z_j}\widehat{\mu}^{s'_i,\pi_Z(s'_i)}(\Delta_j)/\Delta\log q_\theta^{s_i,a_i}(\Delta_j)/\Delta\right)$$

$$=\frac{1}{n}\sum_{i=1}^{n}\left((1-\epsilon)(-\log q_\theta^{s_i,a_i}(\Delta_E^i)/\Delta)-\epsilon\sum_{j=1}^{N}\widehat{\mu}^{s'_i,\pi_Z(s'_i)}(\Delta_j)\log q_\theta^{s_i,a_i}(\Delta_j)/\Delta\right)$$

$$\propto\frac{1}{n}\sum_{i=1}^{n}\left((1-\epsilon)(-\log q_\theta^{s_i,a_i}(\Delta_E^i))+\epsilon\mathcal{H}(\widehat{\mu}^{s'_i,\pi_Z(s'_i)}, q_\theta^{s_i,a_i})\right)$$

$$\propto\frac{1}{n}\sum_{i=1}^{n}\left(-\log q_\theta^{s_i,a_i}(\Delta_E^i)+\alpha\mathcal{H}(\widehat{\mu}^{s'_i,\pi_Z(s'_i)}, q_\theta^{s_i,a_i})\right),\text{ where }\alpha=\frac{\epsilon}{1-\epsilon}>0$$

$$(27)$$

where the cross entropy $\mathcal{H}(\widehat{\mu}^{s'_i,\pi_Z(s'_i)}, q_\theta^{s_i,a_i})$ is based on the discrete distribution when $i=1,...,N$. $\Delta_E^i$ represent the interval that $\mathbb{E}\left[Z^\pi(s'_i,\pi_Z(s'_i))\right]$ falls into, i.e., $\mathbb{E}\left[Z^\pi(s'_i,\pi_Z(s'_i))\right]\in\Delta_E^i$. $\qquad\square$

## G  PROOF OF PROPOSITION 4

**Proposition 4** In Eq. 2 of Neural FZI, if the function class $\{Z_\theta:\theta\in\Theta\}$ is sufficiently large such that it contains $\{Y_i\}_{i=1}^{n}$, as $\Delta\to 0$ ($N\to+\infty$), we have

$$P(Z_\theta^{k+1}(s,a)=\mathcal{T}^{\text{opt}}Q_{\theta^*}^k(s,a))=1,\qquad(28)$$

where $\mathcal{T}^{\text{opt}}Q_{\theta^*}^k(s,a)$ is the target in Eq. 1 of Neural FQI.

*Proof.* Firstly, we define the distributional Bellman optimality operator $\mathfrak{T}^{\text{opt}}$ as follows:

$$\begin{aligned}\mathfrak{T}^{\text{opt}}Z(s,a)&\overset{D}{=}R(s,a)+\gamma Z\left(S',a^*\right)\\ S'&\sim P(\cdot\mid s,a),\quad a^*=\underset{a'}{\operatorname{argmax}}\mathbb{E}\left[Z\left(S',a'\right)\right]\end{aligned}\qquad(29)$$

If $\{Z_\theta:\theta\in\Theta\}$ is sufficiently large enough such that it contains $\mathfrak{T}^{\text{opt}}Z_{\theta^*}$, then optimizing Neural FZI in Eq. 2 leads to $Z_\theta^{k+1}=\mathfrak{T}^{\text{opt}}Z_{\theta^*}$.

We apply the action-value density function decomposition on the target histogram function $\widehat{p}^{s,a}(x)$. Consider the parameterized histogram density function $h_\theta$ and denote $h_\theta^E/\Delta$ as the bin height in the bin $\Delta_E$, under the KL divergence between the first histogram function $\mathbb{1}(x\in\Delta_E)$ with $h_\theta(x)$, the objective function is simplified as

$$\begin{aligned}D_{\text{KL}}(\mathbb{1}(x\in\Delta_E)/\Delta, h_\theta(x))&\propto-\int_{x\in\Delta_E}\frac{1}{\Delta}\log\frac{h_\theta^E}{\Delta}dx\\ &\propto-\log h_\theta^E\end{aligned}\qquad(30)$$

Since $\{Z_\theta:\theta\in\Theta\}$ is sufficiently large enough, the KL minimizer would be $\widehat{h}_\theta=\mathbb{1}(x\in\Delta_E)/\Delta$ in expectation. Then, $\arg\min_{h_\theta}\lim_{\Delta\to 0}D_{\text{KL}}(\mathbb{1}(x\in\Delta_E)/\Delta, h_\theta(x))=\delta_{\mathbb{E}[Z^{\text{target}}(s,a)]}$, where $\delta_{\mathbb{E}[Z^{\text{target}}(s,a)]}$ is a Dirac Delta function centered at $\mathbb{E}\left[Z^{\text{target}}(s,a)\right]$ and can be viewed as a generalized probability density function. The limit behavior from a histogram function $\widehat{p}$ to a continuous one for $Z^{\text{target}}$ is guaranteed by Theorem 1, and this also applies from $h_\theta$ to $Z_\theta$. In Neural FZI, we have $Z^{\text{target}}=\mathfrak{T}^{\text{opt}}Z_{\theta^*}$. According to the definition of Dirac function, as $\Delta\to 0$, we attain

$$P(Z_\theta^{k+1}(s,a)=\mathbb{E}\left[\mathfrak{T}^{\text{opt}}Z_{\theta^*}^k(s,a)\right])=1\qquad(31)$$

Due to the linearity of expectation analyzed in Lemma 4 of (Bellemare et al., 2017a), we have

$$
\begin{aligned}
\mathbb{E}\left[\mathfrak{T}^{\mathrm{opt}} Z_{\theta^*}^k(s,a)\right] &= \mathfrak{T}^{\mathrm{opt}} \mathbb{E}\left[Z_{\theta^*}^k(s,a)\right] \\
&= \mathcal{T}^{\mathrm{opt}} Q_{\theta^*}^k(s,a)
\end{aligned}
\tag{32}
$$

Finally, we obtain:

$$
P(Z_\theta^{k+1}(s,a) = \mathcal{T}^{\mathrm{opt}} Q_{\theta^*}^k(s,a)) = 1 \quad \text{as } \Delta \to 0 \tag{33}
$$

$\square$

# H PROOF OF CONVERGENCE OF SOFT DISTRIBUTIONAL POLICY ITERATION IN THEOREM 2

## H.1 PROOF OF SOFT DISTRIBUTIONAL POLICY EVALUATION IN LEMMA 1

**Lemma 1**(Distribution-Entropy-Regularized Policy Evaluation) Consider the distribution-entropy-regularized Bellman operator $\mathcal{T}_d^\pi$ in Eq. 8 and the behavior of expectation of $Z^\pi(s,a)$, i.e., $Q(s,a)$. Assume $\mathcal{H}(\mu^{s_t,a_t}, q_\theta^{s_t,a_t}) \leq M$ for all $(s_t,a_t) \in \mathcal{S} \times \mathcal{A}$, where $M$ is a constant. Define $Q^{k+1} = \mathcal{T}_d^\pi Q^k$, then $Q^{k+1}$ will converge to a *corrected* Q-value of $\pi$ as $k \to \infty$ with the new objective function defined as

$$
J'(\pi) = \sum_{t=0}^T \mathbb{E}_{(s_t,a_t)\sim\rho_\pi}\left[r(s_t,a_t) + \gamma f(\mathcal{H}(\mu^{s_t,a_t}, q_\theta^{s_t,a_t}))\right]. \tag{34}
$$

*Proof.* Firstly, we plug in $V(s_{t+1})$ into RHS of the iteration in Eq. 8, then we obtain

$$
\begin{aligned}
&\mathcal{T}_d^\pi Q(s_t, a_t) \\
&= r(s_t,a_t) + \gamma \mathbb{E}_{s_{t+1}\sim P(\cdot|s_t,a_t)}[V(s_{t+1})] \\
&= r(s_t,a_t) + \gamma f(\mathcal{H}(\mu^{s_t,a_t}, q_\theta^{s_t,a_t})) + \gamma \mathbb{E}_{(s_{t+1},a_{t+1})\sim\rho^\pi}[Q(s_{t+1},a_{t+1})] \\
&\triangleq r_\pi(s_t,a_t) + \gamma \mathbb{E}_{(s_{t+1},a_{t+1})\sim\rho^\pi}[Q(s_{t+1},a_{t+1})],
\end{aligned}
\tag{35}
$$

where $r_\pi(s_t,a_t) \triangleq r(s_t,a_t) + \gamma f(\mathcal{H}(\mu^{s_t,a_t}, q_\theta^{s_t,a_t}))$ is the entropy augmented reward we redefine. Applying the standard convergence results for policy evaluation (Sutton & Barto, 2018), we can attain that this Bellman updating under $\mathcal{T}_d^\pi$ is convergent under the assumption of $|\mathcal{A}| < \infty$ and bounded entropy augmented rewards $r_\pi$. $\square$

## H.2 POLICY IMPROVEMENT WITH PROOF

**Lemma 2.** *(Distribution-Entropy-Regularized Policy Improvement) Let $\pi \in \Pi$ and a new policy $\pi_{new}$ be updated via the policy improvement step in Eq. 11. Then $Q^{\pi_{new}}(s_t,a_t) \geq Q^{\pi_{old}}(s_t,a_t)$ for all $(s_t,a_t) \in \mathcal{S} \times \mathcal{A}$ with $|\mathcal{A}| \leq \infty$.*

*Proof.* The policy improvement in Lemma 2 implies that $\mathbb{E}_{a_t\sim\pi_{new}}[Q^{\pi_{old}}(s_t,a_t)] \geq \mathbb{E}_{a_t\sim\pi_{old}}[Q^{\pi_{old}}(s_t,a_t)]$, we consider the Bellman equation via the distribution-entropy-regularized Bellman operator $\mathcal{T}_{sd}^\pi$:

$$
\begin{aligned}
Q^{\pi_{old}}(s_t,a_t) &\triangleq r(s_t,a_t) + \gamma \mathbb{E}_{s_{t+1}\sim\rho}[V^{\pi_{old}}(s_{t+1})] \\
&= r(s_t,a_t) + \gamma f(\mathcal{H}(\mu^{s_t,a_t}, q_\theta^{s_t,a_t})) + \gamma \mathbb{E}_{(s_{t+1},a_{t+1})\sim\rho^{\pi_{old}}}[Q^{\pi_{old}}(s_{t+1},a_{t+1})] \\
&\leq r(s_t,a_t) + \gamma f(\mathcal{H}(\mu^{s_t,a_t}, q_\theta^{s_t,a_t})) + \gamma \mathbb{E}_{(s_{t+1},a_{t+1})\sim\rho^{\pi_{new}}}[Q^{\pi_{old}}(s_{t+1},a_{t+1})] \\
&= r_{\pi_{new}}(s_t,a_t) + \gamma \mathbb{E}_{(s_{t+1},a_{t+1})\sim\rho^{\pi_{new}}}[Q^{\pi_{old}}(s_{t+1},a_{t+1})] \\
&\vdots \\
&\leq Q^{\pi_{new}}(s_{t+1},a_{t+1}),
\end{aligned}
\tag{36}
$$

where we have repeated expanded $Q^{\pi_{old}}$ on the RHS by applying the distribution-entropy-regularized distributional Bellman operator. Convergence to $Q^{\pi_{new}}$ follows from Lemma 1. $\square$

### H.3 PROOF OF SOFT DISTRIBUTIONAL POLICY ITERATION IN THEOREM 2

**Theorem 2** (Distribution-Entropy-Regularized Policy Iteration) Assume $\mathcal{H}(\mu^{s_t,a_t}, q_\theta^{s_t,a_t}) \leq M$ for all $(s_t, a_t) \in \mathcal{S} \times \mathcal{A}$, where $M$ is a constant. Repeatedly applying distribution-entropy-regularized policy evaluation in Eq. 8 and the policy improvement in Eq. 11, the policy converges to an optimal policy $\pi^*$ such that $Q^{\pi^*}(s_t, a_t) \geq Q^\pi(s_t, a_t)$ for all $\pi \in \Pi$.

*Proof.* The proof is similar to soft policy iteration (Haarnoja et al., 2018). For the completeness, we provide the proof here. By Lemma 2, as the number of iteration increases, the sequence $Q^{\pi_i}$ at $i$-th iteration is monotonically increasing. Since we assume the risk-sensitive entropy is bounded by $M$, the $Q^\pi$ is thus bounded as the rewards are bounded. Hence, the sequence will converge to some $\pi^*$. Further, we prove that $\pi^*$ is in fact optimal. At the convergence point, for all $\pi \in \Pi$, it must be case that:

$$\mathbb{E}_{a_t \sim \pi^*}\left[Q^{\pi_{\text{old}}}(s_t, a_t)\right] \geq \mathbb{E}_{a_t \sim \pi}\left[Q^{\pi_{\text{old}}}(s_t, a_t)\right].$$

According to the proof in Lemma 2, we can attain $Q^{\pi^*}(s_t, a_t) > Q^\pi(s_t, a_t)$ for $(s_t, a_t)$. That is to say, the "corrected" value function of any other policy in $\Pi$ is lower than the converged policy, indicating that $\pi^*$ is optimal. □

## I IMPLEMENTATION DETAILS

Our implementation is directly adapted from the source code in (Ma et al., 2020).

For Distributional SAC with C51, we use 51 atoms similar to the C51 (Bellemare et al., 2017a). For distributional SAC with quantile regression, instead of using fixed quantiles in QR-DQN, we

Table 1: Hyper-parameters Sheet.

| Hyperparameter | Value |
|---|---|
| *Shared* | |
| Policy network learning rate | 3e-4 |
| (Quantile) Value network learning rate | 3e-4 |
| Optimization | Adam |
| Discount factor | 0.99 |
| Target smoothing | 5e-3 |
| Batch size | 256 |
| Replay buffer size | 1e6 |
| Minimum steps before training | 1e4 |
| *DSAC with C51* | |
| Number of Atoms ($N$) | 51 |
| *DSAC with IQN* | |
| Number of quantile fractions ($N$) | 32 |
| Quantile fraction embedding size | 64 |
| Huber regression threshold | 1 |

| Hyperparameter | Temperature Parameter $\beta$ | Max episode lenght |
|---|---|---|
| Walker2d-v2 | 0.2 | 1000 |
| Swimmer-v2 | 0.2 | 1000 |
| Reacher-v2 | 0.2 | 1000 |
| Ant-v2 | 0.2 | 1000 |
| HalfCheetah-v2 | 0.2 | 1000 |
| Humanoid-v2 | 0.05 | 1000 |
| HumanoidStandup-v2 | 0.05 | 1000 |
| BipedalWalkerHardcore-v2 | 0.002 | 2000 |

leverage the quantile fraction generation based on IQN (Dabney et al., 2018a) that uniformly samples quantile fractions in order to approximate the full quantile function. In particular, we fix the number of quantile fractions as $N$ and keep them in an ascending order. Besides, we adapt the sampling as $\tau_0 = 0, \tau_i = \epsilon_i / \sum_{i=0}^{N-1}$, where $\epsilon_i \in U[0,1], i = 1, ..., N$.

### I.1 HYPER-PARAMETERS AND NETWORK STRUCTURE.

We adopt the same hyper-parameters, which is listed in Table 1 and network structure as in the original distributional SAC paper (Ma et al., 2020).

## J DERAC ALGORITHM

---
**Algorithm 1** Distribution-Entropy-Regularized Actor Critic (DERAC) Algorithm
---
1: Initialize two value networks $q_\theta$, $q_{\theta^*}$, and policy network $\pi_\phi$.
2: **for** each iteration **do**
3:     **for** each environment step **do**
4:         $a_t \sim \pi_\phi(a_t|s_t)$.
5:         $s_{t+1} \sim p(s_{t+1}|s_t, a_t)$.
6:         $\mathcal{D} \leftarrow \mathcal{D} \cup \{(s_t, a_t, r(s_t, a_t), s_{t+1})\}$
7:     **end for**
8:     **for** each gradient step **do**
9:         $\theta \leftarrow \theta - \lambda_q \nabla_\theta \hat{J}_q(\theta)$
10:        $\phi \leftarrow \phi + \lambda_\pi \nabla_\phi \hat{J}_\pi(\phi)$.
11:        $\theta^* \leftarrow \tau\theta + (1-\tau)\theta^*$
12:     **end for**
13: **end for**
---

## K EXPERIMENTS

Figure 5 suggests that C51 with cross entropy loss behaves similarly to the vanilla C51 equipped with KL divergence.

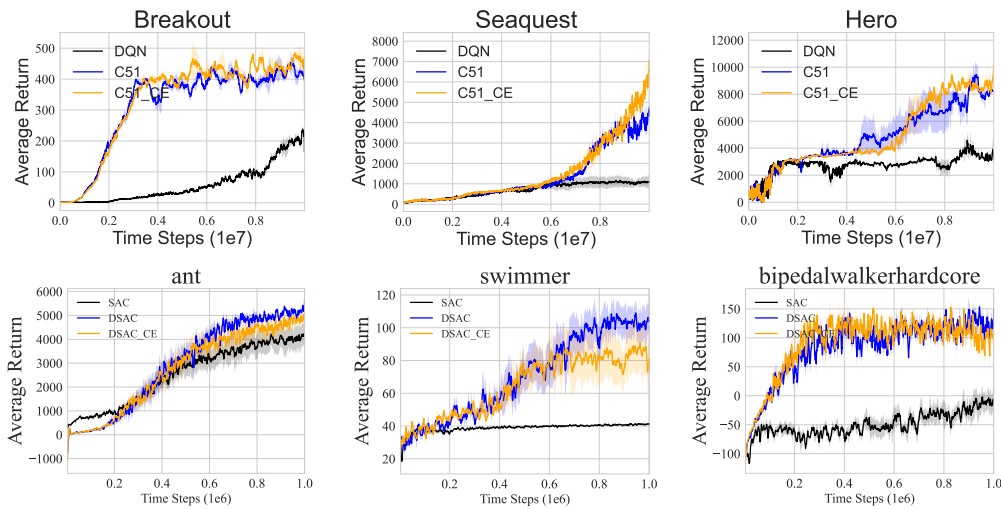

Figure 5: (**First row**) Learning curves of C51 under cross entropy loss on Atari games over 3 seeds. (**Second row**) Learning curves of DSAC with C51 under cross entropy loss on MuJoCo environments over 5 seeds.

Figure 6 shows that DERAC with different $\lambda$ in Eq. 12 may behave differently on the different environment. Learning curves of DERAC with an increasing $\lambda$ will tend to DSAC (C51), e.g.,

Bipedalwalkerhardcore, where DERAC with $\lambda = 1$ in the green line tends to DSAC (C51) in the blue line. However, DERAC with a small $\lambda$ is likely to outperform DSAC (C51) by only leverage the expectation effect of value distribution, e.g., on Bipedalwalkerhardcore, where DERAC with $\lambda = 0, 0.5$ bypass DERAC with $\lambda = 1.0$.

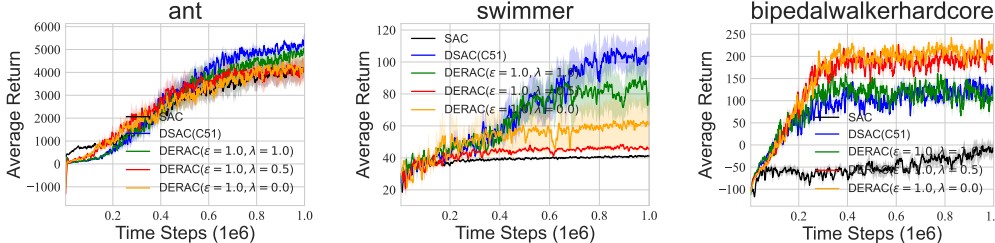

Figure 6: Learning curves of DERAC algorithms across different $\lambda$ on three MuJoCo environments over 5 seeds.

