# OpenReview forum: "Interpreting Distributional Reinforcement Learning: A Regularization Perspective"
_ICLR.cc/2023/Conference — Submitted to ICLR 2023_

### Official Review · Reviewer_UayF · 2022-10-24

**Confidence:** 4
**Correctness:** 3
**Technical Novelty And Significance:** 3
**Empirical Novelty And Significance:** 3
**Recommendation:** 3

**Clarity, Quality, Novelty And Reproducibility:**

This paper's results do appear to be novel and significant, modulo the question regarding the decomposition assumption I mentioned above.

The main issue for me is the clarity, which needs substantial improvement.

**Strength And Weaknesses:**

# Strengths
This paper presents an interesting analysis of distributional RL that can be useful for better understanding its differences with traditional (expectational) RL, and likely lead to better algorithms.

# Weaknesses
## Clarity
The main weakness of this paper is its clarity. There is a lot going on and lots of notation to keep track of (sometimes not presented consistently) which makes it hard to follow. Some points related to clarity below:
1. What is $\mu$ in equation (4)? It has not been introduced yet.
1. In the discussion after equation (4) up to the end of page 3 it's not very clear what $F^{s,a}$ and $F^{s,a}_{\mu}$ are, how they differ, and what the role of each is. A more elaborate discussion may help address point 1 above.
1. At the bottom of page 3 it says "Next, we use $p^{s,a}(x)$ to express the true target probability...", but then it doesn't seem like $p^{s,a}$ is used again. Should this be $\mu^{s,a}$?
1. In the discussion below equation (6) it's interesting and somewhat counterintuitive that estimates at the current $s,a$ are evaluated against the next state-action $s',a'$. It would be nice to have some more discussion, and perhaps even a diagram/figure for better exposing this.
1. In equation (7), where did the horizon $T$ come from? This is the first time it's been introduced and seems to be in conflict with the discounted and infinite-horizon problem setup that has been introduced thus far.
1. In equation (8) should the expectation be sampling from $P(\cdot | s_t, a_t)$ instead of $\rho^{\pi}$?
1. Equation (9) seems to me to be somewhat in conflict with what was presented in equation (6), where estimates at the current $s,a$ are evaluated against the next state-action $s',a'$. The rest of the discussion seems to follow equation (9), but then I'm confused as to how equation (6) (and the discussion following it) fits in.
1. In equation (9) (and sentence following it) it seems we're in a tabular setting, yet you have $q_{\theta}$. What is the role of $q_{\theta}$ in a tabular scenario? What is it approximating and how?
1. In the paragraph following equation (9) it says "can normally be obtained via bootstrap estimate $\mu^{s_{t+1},\pi_Z(s_{t+1})}$ similar in Eq. 6." Can you elaborate on this? It's not clear what is meant by this statement.
1. In Lemma 1 and Theorem 1 there is an assumption of an upper-bound $M$ on the entropy term. Can this be any constant? From the proof it seems all that is necessary is boundedness, so it would be good to clarify the role of $M$ here. In particular, for finite states and actions the entropy term will always be bounded, so the assumption seems to be only necessary when one or the other is not finite.
1. More importantly, shouldn't the upper-boundedness assumption be on $f(\mathcal{H}(\ldots))$? Otherwise, it seems one can pick $f$ adversarially to make the theorem false.
1. In Lemma 1, what is the $sd$ subscript in one of the $\mathcal{T}$s? Should it just be $\mathcal{T}_d$?
1. Why are you using a finite horizon $T$ in the statement of Lemma 1? It does not appear to be necessary and is related to the point made above.
1. Figure 1 is nice, but it's not totally clear what the authors are trying to show there. Providing more details in the caption (or in the main text) would help illustrate how it connects to the paper's analyses.
1. In the first line of page 7 is the $\epsilon$ with $\mathbb{E}[\epsilon]=0$ the same $\epsilon$ from equation (4)?
1. In the line after equation (12) it says "we consider a particular increasing function $f(\mathcal{H})$, but there is no $f$ in equation (12).
1. In the first paragraph of section 4 it says QR-DQN is also evaluated, but then later it seems the authors use IQN, not QR-DQN.
1. In section 4.1 there is a sentence "Firstly, it is a fact that the value distribution decomposition... owing to the leverage of target networks." It is not clear what is meant by this sentence.
1. In section 4.1 it says "demonstrate that C51 algorithm can still achieve similar results _under the cross entropy loss_", which is a little confusing, since C51 was introduced with a cross entropy loss.
1. In the first paragraph of section 4.1 there is some discussion using $\varepsilon$ which is rather confusing:
    1. Is it same as the $\epsilon$ used earlier in the paragraph (for the continuous decomposition) or different?
    1. The same $\varepsilon$ is used to refer to "the proportion probability" _and_ "the true $\varepsilon$", which makes things confusing.
    1. It appears $\varepsilon$ is referring to the same $\epsilon$ object from equation (4), but they are different symbols, which makes things confusing.
1. In general, I found the first paragraph of section 4.1 really hard to follow. Since it is quite important to the paper's contributions, it needs to be clarified.

## Assumptions
In section 3.2 the authors assume that $F^{s,a}$ satisfies ... eqn (4). The rest of the paper's analysis seems to hinge on this assumption, but it is not clear to me how reasonable an assumption it is to make.

## Comparison to related work
Another weakness is that it is missing discussion with a very related work:
* [Clare Lyle, Pablo Samuel Castro, Marc G. Bellemare. A Comparative Analysis of Expected and Distributional Reinforcement Learning. AAAI 2019](https://arxiv.org/abs/1901.11084)

Lyle et al. prove that for tabular and linear function approximation, distributional and expectational RL are equivalent; with non-linear approximators distributional and expectational are _different_, but sometimes distributional can hurt performance. This seems to be in contrast with some of the wording in the paper under review, where the authors make statements like (emphasis is mine): "we illuminate the **superiority** of distributional RL over expectation-based RL.".

# Minor suggestions
* Second line of page 2 should read: "result is based on two analy**tical** components, ... by leverag**ing a** variant of gross..."
* Top line of page 5 should say "risk-neu**t**ral".
* In section 3.3 when you introduce _Distribution-Entropy-Regularized Policy Iteration_ you may as well introduce the acronym DERPI (instead of waiting until the next page).
* Above equation (12) it should say "**objective** function", not "objection function".
* Below equation (12) it says "we use the target value distribution neural network $q_{\theta^*}$". Then why not just use $q_{\theta^*}$ directly in equation (12) to make it clear?
* In **Environments** in section 4 it says three Atari games were used: Breakout, Seaquest, and Asterix. But it seems Hero is being used instead of Asterix.
* Please specify what the shaded areas represent in your figures.
* Please restate lemmas/theorems in the appendix when you present the proofs.

**Summary Of The Paper:**

This paper performs an analysis of distributional reinforcement learning using a regularization perspective. Specifically, under the assumption of a particular decomposition of the value distribution, they argue that distributional RL provides a risk-sensitive entropy regularization when approximated via the neural fitted Z-iteration framework (something also introduced in this paper).

**Summary Of The Review:**

I think this is a nice paper that can be an important contribution to further understanding distributional reinforcement learning and can help advance research in this area. However, there is a fair bit of work to do on the clarity front to make this paper useful for others wishing to build on it.

As it is, I do not believe this paper is ready for publication yet. However, I do believe the authors have an important contribution to provide, so I would encourage them to work on the clarity of the writing and exposition.

---

> ### Author Response · Authors · 2022-11-17
> **Author Reponse**
>
> Thank you for your valuable comments and suggestions. Below are our responses to your concerns. Please be free to let us know if there are any further questions.
>
> ### Clarity.
> * 1. $\mu$ is the resulting probability density function after the distribution decomposition.
> * 2. We emphasized it.
> * 3. $p^{s,a}$ is used in the proof. We delete this sentence.
> * 4. The evaluation based on the next state action pair is based on TD(0) learning.
> * 5. Similar to SAC algorithm, we consider the episodic setting. In infinite-horizon/continuing problem, we normally use the average reward as the objective function.
> * 6. We revised it.
> * 7. Eq(6) is directly based on the distribution Bellman Optimality operator, and thus we directly apply $(s^\prime, a^\prime)$. In Eq.(9), $\mu^{s, a}$ is the target of $q_\theta^{s, a}$. We can definitely use distribution TD estimate $\mu^{s^\prime, a^\prime}$ to approximate $\mu^{s, a}$.
> * 8. We can view it as a ``distributional’’ tabular setting, where in each s, a, we need to compute a value distribution rather than a * single Q value. This new setting, albeit straightforward, extends to the vanilla Q values-based tabular setting.
> * 9. Please refer to ‘Remark on the attainability of $\mu(s^\prime, a^\prime)$’. In principle, it can be evaluated via distributional TD learning.
> * 10. The boundness assumption guarantees the convergence of the Q function. Please refer to Appendix H.3 for more details.
> * 11. It might be. As in our DERAC algorithm, we choose a particular function $f$ and it does not involve this issue. However, in the general case, the upper-boundness property of $f$ may be useful for other theoretical analyses, which we leave as future works.
> * 12. We revised it.
> * 13. We followed the SAC paper and consider the episodic setting.
> * 14. We revised it.
> * 15. We revised it.
> * 16. We revised it.
> * 17. We revised it.
> * 18. We revised the action-value density decomposition, which makes the explanation of this part consistent with the theoretical analysis in Section 3.2 in the revised paper.
> * 19. C51 uses KL divergence, and we show that by leveraging the target network, KL divergence is equivalent to cross-entropy. Thus, C51 with cross-entropy behaves similarly to vanilla C51.
> * 20. We revised it. We use $\epsilon$ in the decomposition and $\varepsilon$ as the surrogate. Please refer to Section 4.1 for the revised explanation.
> * 21. We revised it.
>
> ### Assumption.
> We thank you for raising this insightful question. To address this limitation of a restrained distribution function class, we revised the action-value distribution decomposition from the continuous form to the discrete one via the histogram density estimate, which can **approximate arbitrarily continuous action-value distribution**. We also provide the approximation analysis (Theorem 1 in the revised version) as well as the sufficient and necessary conditions (Proposition 1 in the revised version) to guarantee the decomposition is valid. This new decomposition as well as the revised explanation can **fundamentally address the restrained distribution class issue** in the original paper version. We invite you to have a detailed look at the revised paper, and we hope this revision could help to eliminate your concern.
>
> ### Comparison to Related works.
> Thanks for providing this reference. Based on your suggestion, we add the discussion of this paper in the introduction part. This paper proved in many realizations of tabular and linear approximation settings, distributional RL behaves the same as expectation-based RL **under the coupling updates method**, but diverges in non-linear approximation. By contrast, our work mainly focuses on the **non-linear approximation setting** and explains the behavior difference of distributional into its regularization effect, which is from a different perspective and provides more insights.
>
> ### Minor Suggestions.
> We revised these issues. Shaded areas represent the confidence interval in our experiments.

---

> > ### Comment · Reviewer_UayF · 2022-11-18
> > **Latest revisions**
> >
> > Thank you for responding to my comments and for your revised submission. The changes to the paper are quite substantial... more than can be properly reviewed using the UI of OpenReview's Reivision Comparison tool. Thus, I will ahve to go through the changes manually, but this unfortunately will take me a bit more time (unlikely I'll be able to finish before the end of the discussion period with authors).
> >
> > Nevertheless, I will take your responses and changes into consideration during the non-author-visible discussion period, and consider revising my score, if appropriate.

---

> > > ### Author Response · Authors · 2022-11-18
> > > **Thank you**
> > >
> > > Sure, we appreciate your time and response. The main revision is based on the decomposition assumption in Section 3.2 from the original continuous decomposition to a discrete one, along with further rigorous proof. It is still within the scope of our submission paper and experimental results and conclusions are also the same. Looking forward to the discussion during the non-author-visible discussion period.

---

### Official Review · Reviewer_RieR · 2022-10-25

**Confidence:** 4
**Correctness:** 3
**Technical Novelty And Significance:** 2
**Empirical Novelty And Significance:** 1
**Recommendation:** 3

**Clarity, Quality, Novelty And Reproducibility:**

The paper studies an important topic in RL, however, the writing lacks some clarity. In particular the italics in many places makes it hard to read at times. The paper is original and may have impact if the empirical validation could be shown.

**Strength And Weaknesses:**

Strengths:
- This paper studies an important topic in distributional RL, by looking at where the superior performance of distributional RL algorithms comes from. It further presents a new framework by incorporating an entropy term of the value-distribution itself (as opposed to the policy entropy) into the Q-value estimation. Furthermore, it retrofits the analysis into the policy iteration and policy evaluation frameworks, which is a familiar theoretical framework.
- The proposed entropy regularization term is compatible with existing distributional RL algorithms, in particular, the quantile based ones and for both discrete/continous control.

Weaknesses:
- The theoretical contributions, while interesting are performed on a KL divergence as a distance measure between distributional value-estimates. While the authors acknowledge this is done because Wasserstein distance is less manageable in this theoretical framework, the KL divergence is non-expanding while most state-of-the-art distributional RL papers have moved to contractive distance measures such as Wasserstein or MMD. This discrepancy would seem to lessen the contribution of this paper.
- The experimental results are not convincing beyond to study the effects of regularizing the update using the proposed DERAC algorithm. The DSAC and C51 comparisons use "vanilla entropy regularization" and no regularization and outperfoms DERAC on a majority of tasks. I understand the authors state the empircal results are to corroborate the theoretical results and not necessarily to perform well against existing algorithms, but then it seems there should be additional/different experiments analyzing how exactly the it converges to the optimal risk-sensitive policy.
- The experiments given in Figure 4. are confusing in terms of drawing conclusions, since they combine two different frameworks for RL. Notably, it is unclear how VE and RE tradeoff against one another.

**Summary Of The Paper:**

his paper addresses the lack of theoretical analysis on why distributional RL works so well and approaches it from the framework of risk-sensitive entropy regularization. It introduces an entropy term in the distributional bellman update which is different from that of conventional MaxEnt frameworks, as it is an entropy of the value-distribution and is state-action-wise rather than just state-wise.


**Summary Of The Review:**

I think this paper requires additional polish and refinement, in particular some empirical demonstrations of the usefulness of the theoretical analyses made. I would lean towards rejecting.

---

> ### Author Response · Authors · 2022-11-17
> **Author Response**
>
> Thank you for your valuable comments and suggestions. Below are our responses to your concerns. Please be free to let us know if there are any further questions.
>
> ### Weakness 1. Non-expansive KL divergence.
> The KL divergence is a typical choice in categorical distributional RL, which can be viewed as the first successful distributional RL family. In categorical distributional RL, a projection $\Pi_{\mathcal{C}}$ equipped with KL divergence is proven to be convergent under Cramer distance[1], which possesses many of the same properties as the Wasserstein metric[2]. Based on your suggestion, we add a remark paragraph at the head of Page 5 in the current version. **Our target action-value distribution decomposition is applied after the projection $\Pi_{\mathcal{C}}$, when equipped with KL divergence, the distributional Bellman operator can still be guaranteed to be contractive under Cramer distance[1]**.
>
> Clearly, it is not perfect as our analysis is largely based on categorical distribution equipped with KL divergence, it is definitely the first step and more theoretically manageable than the Wasserstein distance. We leave analysis on Wasserstein distance as future works.
>
> ### Weakness 2. Experimental Results
> We think both you and us agree on our empirical demonstration point as we mainly focus on verifying the convergence of DERAC guaranteed by the theoretical convergence of DERPI, rather than showing its superiority. We also mentioned this point in Section 4.2. **In practice, we can directly apply the distributional RL algorithm to pursue better performance**, while experiments about DERAC are to demonstrate the distributional regularization effect arising from distributional RL.
>
> For the risk-sensitive policy, it is in fact an open problem to investigate the optimal one empirically as our risk-sensitive regularization corresponds to a neutral risk analyzed at the end of Page 4. **The optimal risk-sensitive policy highly depends on the environment in nature**, which is very complicated to dive deeper into. Any further suggestions from you would be much appreciated.
>
> ### Weakness 3. Different Exploration Strategies.
> SAC encourages the policy to visit states with high entropy to pursue the diversity of states to optimize, while distributional RL promotes the risk-sensitive exploration to visit state and action pairs whose action-value distribution has a larger degree of dispersion. We hypothesize that **mixing two different exploration directions may lead to sub-optimal solutions in certain environments, thus interfering with each other eventually**.
>
> [1] An Analysis of Categorical Distributional Reinforcement Learning (AISTATS 2018)
> [2] The Cramer Distance as a Solution to Biased Wasserstein Gradients (NeurIPS 2017)

---

> ### Comment · Area_Chair_NjRx · 2022-11-20
> **Any comments to the responses from authors?**
>
> Dear Reviewer RieR,
>
> Thank you very much for your informative review.  The authors have provided responses to your concerns.  Did they resolve your concerns?  How did they change your evaluation?

---

### Official Review · Reviewer_wnYb · 2022-11-02

**Confidence:** 4
**Correctness:** 2
**Technical Novelty And Significance:** 3
**Empirical Novelty And Significance:** 2
**Recommendation:** 3

**Clarity, Quality, Novelty And Reproducibility:**

- For clarity, quality, and novelty, please see the comments above.
- For reproducibility, the experiments in this paper are built on the source code of DSAC, and the configuration is provided in the appendix. The experiments shall be reproducible.


**Strength And Weaknesses:**

**Strength**

- A new perspective of interpreting distributional RL: This paper aims to answer an important question in the distributional RL literature: why do the distributional RL methods have an empirical advantage over the expectation-based RL methods? This paper proposes to decompose the objective function into an expectation term and an additional regularization term. As far as I know, this perspective of distributional RL is quite novel.
- This paper is easy to understand for most parts of the paper.
- The proposed method enjoys one additional flexibility (compared to the standard C51) – the weight of value distribution decomposition for controlling the risk sensitivity. This feature could be useful in practice.

**Weaknesses**
- There are some concerns regarding the decomposition assumption of $F^{s, a}$ in Eq. (4): Given an arbitrary CDF $F^{s,a}$, it is not always true that the resulting $F_{\mu}^{s,a}$ is also a valid CDF. This is a critical issue since in the proof of Proposition 3, the cross entropy term in the 4th line of Eq. (19) would not always be well-defined. In other words, to establish Proposition 3, one would need to first show that $\mu$ is indeed a valid PDF. More justification about this assumption is needed.
- The novelty of the proposed algorithm: The proposed DERAC method can be viewed as a variant of C51 (for discrete actions) and DSAC (for continuous actions) with a slightly different regularization term. It would be good to clarify and highlight the differences between DSAC and C51/DERAC.
- Lack of discussion on the experiment results: Several experimental results do not appear very conclusive.
For example, regarding the result in Figure 4, can the authors explain why the curve of AC+RE+VE performs well in the environments of humanoid and walker2d (despite the conjecture mentioned by the authors)?
Moreover, in Section 4.2, it is mentioned that “Our empirical result in Figure 3 has provided strong evidence to verify our theoretical results.” However, this is not completely clear to me since (i) Theorem 1 suggests “convergence to a global optimum” while Figure 3 can only show convergence to some stationary point, and (ii) the empirical result of DERAC cannot be used to corroborate the theoretical results of DERPI given that DERAC is a learning algorithm while DERPI in Theorem 1 is essentially a planning algorithm.
Since DERAC is a reinterpretation of C51 (which is a distributional method that also uses KL divergence), it is a bit surprising that the performance of DERAC and C51 are rather different in Figure 2.

Another thing to mention is about the generality of the analysis: It appears that the analysis (cf. Propositions 1-3) requires specifically choosing the KL-divergence in the objective function. It it not immediately clear whether the analysis indeed capture the essence of distributional RL and can address other popular distributional distance metrics (e.g., Wasserstein distance). While this might not be a true weakness, it would be helpful to have a discussion on this somewhere in the paper.

**Summary Of The Paper:**

This paper provides a new regularization perspective on analyzing why the distribution RL methods perform better than the expectation-based RL methods empirically. By leveraging the decomposition assumption of the action-value distributions, this paper proposes to express the objective function of NeuralFZI as the expected effect and the distributional regularization effect. Based on this decomposition, the author(s) proposed DESPI, which is a variant of the soft policy iteration for distributional entropy regularization. Accordingly, to substantiate DESPI in more practical RL tasks, this paper presents a learning algorithm called DERAC, which performs well in some complex continuous control tasks.

**Summary Of The Review:**

Overall this paper tackles an important question in distributional RL from a novel perspective. My main concerns are the critical assumption of the value distribution decomposition mentioned above as well as that the experiments look quite inconclusive and require more explanation.

---

> ### Author Response · Authors · 2022-11-17
> **Author Response**
>
> Thank you for your valuable comments and suggestions. Below are our responses for your concerns. Please be free to let us know if there are any further questions.
>
> ### Weakness 1. Explanation of Action-value distribution decomposition.
> We thank you for raising this insightful question. In fact, to guarantee a valid pdf of $\mu$, we need to restrain the distribution class $F^{s, a}$ to be even discontinuous, which may limit our theoretical analysis. To address this issue, we revised the action-value distribution decomposition from the continuous form to the discrete one via the histogram density estimate, **which can approximate arbitrarily continuous action-value distribution** under mild considitions. We also provide the approximation analysis (Theorem 1 in the revised version) as well as the sufficient and necessary conditions (Proposition 1 in the revised version) to guarantee the decomposition is valid. This new decomposition as well as the revised explanation can **fundamentally address the restrained distribution class issue** in the original paper version. We invite you to have a detailed look at the revised paper, and we hope this revision could help to eliminate your concern.
>
> ### Weakness 2. Relationship with DSAC and C51
> Essentially, DERAC extracts the distributional regularization part from C51 distributional loss in Distributional SAC (DSAC) via the action-value distribution decomposition technique we proposed. As explained in the paragraph below Eq.(12), when $\lambda=1$ and we use the whole target distribution $p^{s,a}$ to approximate $\mu^{s,a}$, the critic loss in DERAC would degrade to C51 loss in DSAC. In general, the proposal of DERAC is to further demonstrate the role of risk-sensitive regularization in the actor-critic framework to better connect with maximum entropy RL.
>
> ### Weakness 3. Experimental results.
> **(1)Joint benefit revealed by AC+RE+VE**.
> SAC encourages the policy to visit states with high entropy to pursue the diversity of states to optimize, while distributional RL promotes the risk-sensitive exploration to visit state and action pairs whose action-value distribution has a larger degree of dispersion. Both RE and VE would have joint benefits if their joint exploration behaviors lead to better samples, based on which a better policy can be well expected.
>
> **(2)Convergence discussion.**
> We agree mostly and thus we revised this statement. The theoretical results corroborate the convergence of DERAC (to some stationary points). As DERPI is the tabular version of DERAC with function approximation, results can also provide partial evidence to demonstrate its convergence in the tabular case, although it is not a direct demonstration.
>
> **(3)Surprising difference with C51.**
> This difference results from the sensitivity of DERAC in terms of $\lambda$ in Swimmer. Figure 6 in Appendix K in fact suggests that DERAC with a larger $\lambda$ is very close to DSAC(C51).
>
> ### Weakness 4. The generality of KL divergence
> Thanks for pointing out this issue. We in fact discussed this issue in the final discussion section. In the revised version, we emphasize the **intermediate role of histogram density estimate** we use between categorical distribution in C51 and quantile function to approximate Wasserstein distance. We agree this is a potential limitation as our analysis is highly linked with categorical distribution equipped with KL divergence. Thus, we leave a direct analysis based on the quantile function that approximates Wasserstein distance as future works, although it is theoretically tricky.

---

> ### Comment · Area_Chair_NjRx · 2022-11-20
> **Any comments to the responses from authors?**
>
> Dear Reviewer wnYb,
>
> Thank you very much for your detailed review.  The authors have provided responses to your concerns.  How did they change your evaluation?  In particular, did they response your main concerns on the critical assumption of the value distribution decomposition and inconclusive experiments?

---

### Official Review · Reviewer_8Ekx · 2022-11-03

**Confidence:** 3
**Correctness:** 2
**Technical Novelty And Significance:** 2
**Empirical Novelty And Significance:** 2
**Recommendation:** 5

**Clarity, Quality, Novelty And Reproducibility:**

Clarity:  This paper does not present a clear logic, such as why comparing with Maximum Entropy RL, it is not enough to just say that “establishes a bridge”.

Quality: The quality of this paper is overall well.

Novelty: Novel problem

Reproducibility: No open source code, only provide other people's code


**Strength And Weaknesses:**

Pros
The paper investigates an important problem.
The paper provides solid theoretical analysis.

Cons
1. Many assumptions are made to gain the theoretical results, but the author does not explain them clearly. e.g., (1)In Section 3.2, the author assumes the action value function satisfies the expectation decomposition of Eq.(4). However, the author needs to convince us why it is true. It seems proposition 1 tries to do that, but it is still not clear why action value function satisfies the decomposition.

2. The paper is hard to follow. The theoretical derivation is not explained clearly. For example, in proposition 1, why the inf is taken over F_\mu^{s,a}, also the corresponding prof in Appendix A, what does the “||” mean in the first inequality of Eq.(15)?

3. As the title of the article shows, the main idea of this paper is to explain the superiority of distributional RL. But actually, the author only explains distributional RL that uses KL divergence, and the experiments are also only performed with the C51 that is based on KL divergence. So I think the topic of the paper should be further narrowed down , since the results can not represent all the distributional RL algorithms.

**Summary Of The Paper:**

This paper investigates an important problem of interpreting why distributional RL outperforms conventional RL. Specifically, the author separates the action value function into the expectation part and regularization part, and attributes the superiority to the regularization part. In addition, the author proposes a new algorithm called DERPI for both tabular and function approximation settings. The experimental results manifest the effectiveness.

**Summary Of The Review:**

Generally speaking, this paper aims to address an important problem. However, the paper does not solve the problem very well, since the result can only interpret KL-divengence based distributional RL. In addition, there are also not enough experimental scenarios to support the theoretical results.

---

> ### Author Response · Authors · 2022-11-17
> **Author Response**
>
> Thank you for your valuable comments and suggestions. Below are our responses for your concerns. Please be free to let us know if there are any further questions.
>
> ### Q1. Assumptions
> We appreciate this question. To address it, we revised the action-value distribution decomposition from the continuous form to the discrete one via the histogram density estimate, which can **provably approximate arbitrarily continuous action-value distribution**. We also provide the approximation analysis (Theorem 1 in the revised paper) as well as the sufficient and necessary conditions (Proposition 1 in the revised paper) to guarantee the decomposition is valid. This new decomposition as well as the revised explanation can **fundamentally address the restrained distribution class issue** in the original paper version. We invite you to have a detailed look at the revised paper, and we hope this revision could help to eliminate your concern.
>
> ### Q2. Theoretical Deviation.
> Although in the revised paper this proposition with its proof is deleted as we substitute with a new decomposition, we would like to clarify that the $\inf_{F_\mu}$ is used to measure the minimum distribution difference for $F^{s, a}$ with an arbitrary one regarding a varying $F_\mu$ in $L_\infty$ analysis, and the second $|$ indicates “given a special value”.
>
> ### Q3. Intermediate role of histogram density estimate between a categorical distribution and quantile function.
> It is true that our analysis is largely connected with categorical distributional RL equipped with KL divergence and we also establish the optimization equivalence between the histogram function we use and the categorical distribution in Proposition 2. However, we argue that **the leverage of a histogram density estimate is also linked with quantile function**. We add this discussion at the end of Page 3 of the revised paper. ``Histogram and quantile functions are `two sides of a coin’ The histogram estimates the density function by **giving each bin an equal amount of information**, while the quantile function **gives each fraction of data the same amount of information**. Based on this insight, we thus argue that our analysis based on the histogram density estimate is largely general and representative in distributional RL families’’. We also leave a direct analysis based on Wasserstein distance as future works.

---

> > ### Comment · Reviewer_8Ekx · 2022-11-22
> > **Thanks for The Rebuttal**
> >
> > Dear Authors,
> >
> > Thanks for your rebuttal. In the rebuttal, authors mainly address my concerns about the theoretical results. My other concerns are i) the current flow of the paper is not very clear, which can be largely improved, and ii) more comprehensive experiments can support the claims of the paper. So I would keep my score, given that I gave the highest score, and suggest the authors to improve the current version for further resubmission.

---

> ### Comment · Area_Chair_NjRx · 2022-11-20
> **Any comments to the responses from authors?**
>
> Dear Reviewer 8Ekx,
>
> Thank you very much for your informative review.  The authors have provided responses to your concerns.  How did they change your evaluation?  Did they resolve your concerns?  Do any major concerns still remain?

---

### Author Response · Authors · 2022-11-17
**General Author Reponse**

Dear Reviewers and ACs,

According to reviewers’ suggestions, **we revised the analysis part in Section 3.2 largely**, but it is within the scope of the original paper in submission and maintains the empirical results as well as the conclusions. **We also provide the source code as well as the original paper in submission for reference**. Our revision in the revised paper is in two aspects.

* As most reviewers challenged the limitation of the restrained distribution function class in the action-value distribution decomposition, we revised it into a discrete form based on the histogram density estimation that can **approximate arbitrarily continuous action-value density function** under mild conditions analyzed in Proposition 1. This is also consistent with our empirical implementation as practical algorithms are mainly based on discrete function estimates. We also provide a new Theorem 1 for an approximation analysis.

* We also provide a new Proposition 4 that **connects the first term of decomposition in Neural FZI with Neural FQI**, and thus the resulting regularization term can be more rigorously viewed as the explanation for the behavior difference of distributional RL.

It takes us painstaking efforts to revise our paper accordingly based on reviewers' suggestions, and thus it would be much appreciated if you can take the revision above into your final consideration for our paper. Further responses from reviewers at their convenience would also contribute to the whole research community via this high-profile conference. Thanks a lot.

Yours Sincerely,
Authors

---

### Decision · Program_Chairs · 2023-01-20

**Decision:**

Reject

**Justification For Why Not Higher Score:**

The theoretical contributions are questionable, and experimental support is insufficient.

**Justification For Why Not Lower Score:**

N/A

**Metareview: Summary, Strengths And Weaknesses:**

This paper investigates why distributional reinforcement learning (RL) can outperform conventional expectation-based RL.  The novelty of the analysis of the paper is in the decomposition of action-value distribution.  The reviewers agree that this is an important problem, and the overall approach is novel, which constitute the strength of the paper.

Weaknesses of the paper include insufficiency of experimental support, questionable assumptions about decomposition of the action-value distribution, limited applicability of the theory that depends on KL divergence, and the clarity of writing.